# Metric-Projected Accelerated Riemannian Optimization: Handling Constraints to Bound Geometric Penalties

## Abstract

We propose an accelerated first-order method for the optimization of smooth and (strongly or not) geodesically-convex functions over a compact and geodesically-convex set in Hadamard manifolds, that we access to via a metric-projection oracle. It enjoys the same rates of convergence as Nesterov's accelerated gradient descent, up to a multiplicative geometric penalty and log factors. Even without in-manifold constraints, all prior fully accelerated works require their iterates to remain in some specified compact set (which is needed in worst-case analyses due to a lower bound), while only two previous methods are able to enforce this condition and these, in contrast, have limited applicability, e.g., to local optimization or to spaces of constant curvature. Our results solve an open question in [KY22] and an another question related to one posed in [ZS16]. In our solution, we show we can use projected Riemannian gradient descent to implement an inexact proximal point operator that we use as a subroutine, which is of independent interest.

## 1 Introduction

Riemannian optimization concerns the optimization of a function defined over a Riemannian manifold. It is motivated by constrained problems that can be naturally expressed on Riemannian manifolds allowing to exploit the geometric structure of the problem and effectively transforming it into an unconstrained one. Moreover, there are problems that are not convex in the Euclidean setting, but that when posed as problems over a manifold with the right metric, are convex when restricted to every geodesic, and this allows for fast optimization [Cru+06; CM12; BFO15; All+18]. That is, they are geodesically convex (g-convex) problems, cf. Definition 1.1. Some applications of Riemannian optimization in machine learning include low-rank matrix completion [CA16; HS18; MS14; Tan+14; Van13], dictionary learning [CS17; SQW17], optimization under orthogonality constraints [EAS98; LM19], robust covariance estimation in Gaussian distributions [Wie12], Gaussian mixture models [HS15], operator scaling [All+18], and sparse principal component analysis [GHT15; HW19b; JTU03].

Riemannian optimization, whether under g-convexity or not, is an extensive and active area of research, for which one aspires to develop Riemannian optimization algorithms that share analogous properties to the more broadly studied Euclidean first-order methods, such as the following kinds of Riemannian methods: deterministic [BFM17; Wei+16; ZS16], adaptive [KJM19], projection-free [WS17; WS19], saddle-point-escaping [CB19; SFF19; ZZS18; ZYF19; CB20], stochastic [HS17;

---

[0]Most of the notations in this work have a link to their definitions. For example, if you click or tap on any instance of $L$, you will jump to the place where it is defined as the smoothness constant of the function we consider in this work.

KL17; Tri+18], variance-reduced [SKM17; SKM19; ZRS16], and min-max methods [ZZS22], among others.

Riemannian generalizations to accelerated convex optimization are appealing due to their better convergence rates with respect to unaccelerated methods, specially in ill-conditioned problems. Acceleration in Euclidean convex optimization is a concept that has been broadly explored and has provided many different fast algorithms. A paradigmatic example is Nesterov's Accelerated Gradient Descent (AGD), cf. [Nes83], which can be considered the first general accelerated method, where the conjugate gradients method can be seen as an accelerated predecessor in a more limited scope [Mar21]. There have been recent efforts to better understand this phenomenon in the Euclidean case [AO17; SBC16; DT14; WWJ16; DO19; Jou+20], which have yielded some fruitful techniques for the general development of methods and analyses. These techniques have allowed for a considerable number of new results going beyond the standard oracle model, convexity, or beyond first-order, in a wide variety of settings [Tse08; BT09; WRM16; AO15; All17; All+16; All18b; Car+17; DO18; All18a; CDO18; HSS19; CS19; DJ19; Gas+19; Iva+21; DN20; KG20; CMP21], among many others. There have been some efforts to achieve acceleration for Riemannian algorithms as generalizations of AGD, cf. Section 1.3. These works try to answer the following fundamental question:

*Can a Riemannian first-order method enjoy the same rates of convergence as Euclidean AGD?*

The question is posed under (possibly strongly) geodesic convexity and smoothness of the function to be optimized. And we now know, due to the lower bound in [CB21], that the optimization should be over a bounded domain and under bounded geodesic curvature of the Riemannian manifold. In this work, we study this question in the case of Hadamard manifolds $\mathcal{M}$ of bounded sectional curvature, where many of the applications lie [HS20]. Given a compact and uniquely geodesic g-convex set $\mathcal{X}$ that we access to via a metric-projection oracle, we design first-order algorithms that enjoy the same rates as AGD when approximating $\min_{x \in \mathcal{X}} f(x)$, up to logarithmic factors and up to a geometric penalty factor, where $f : \mathcal{N} \subset \mathcal{M} \to \mathbb{R}$ is a differentiable function that is smooth and g-convex (or strongly g-convex) in $\mathcal{X} \subset \mathcal{N}$. See Section 1.1 for the definitions of these concepts. Importantly, our algorithm obtains acceleration without an undesirable assumption that most previous works had to make: that the iterates of the algorithm stay inside of a specified compact set without any mechanism for enforcing this condition. Only two previous methods are able to deal with some form of constraints, and they apply to the limited settings of constant sectional curvature manifolds and local optimization, respectively. Techniques in the rest of papers can handle neither constraints nor projections, due to fundamental properties of their methods. Removing this condition in general, global, and fully accelerated methods was posed as an open question in [KY22], that we solve for the case of Hadamard manifolds. The difficulty of constraining problems in order to bound geometric penalties as well as the necessity of achieving this goal in order to provide full optimization guarantees is something that has also been noted in other kinds of Riemannian algorithms, cf. [HS20]. See Table 1 for a succint comparison among algorithms with some degree of acceleration and their rates.

The question concerning whether there are Riemannian analogs to Nesterov's algorithm that enjoy similar rates is a question that, to the best of our knowledge, was first formulated in [ZS16]. In particular, since Nesterov's AGD uses a proximal operator of a function's linearization, they ask whether there is a Riemannian analog to this operation that could be used to obtain accelerated rates in the Riemannian case. The natural candidate results in a non-convex problem which is not amenable to optimization. While we do not take this course of action, we show that, instead, a proximal step with respect to the *whole* function can be approximated efficiently in Hadamard manifolds and it can be used along with an accelerated outer loop, when implemented and analyzed carefully, in the spirit of other Euclidean algorithms like Catalyst [LMH17]. It relies on Riemannian gradient descent (RGD) with projections, initialized at a suitable warm-start point that we can find by exploiting the structure of the geometry and the metric projection. The Riemannian proximal point subroutine we design is of independent interest. To the best of our knowledge, previously known Riemannian proximal methods either obtain asymptotic analyses, assume exact proximal computation, or work with approximate proximal operators by using different inexactness conditions as ours, and do not show how to implement the inexact operators, cf. Section 1.3.

## 1.1 Preliminaries

We provide definitions of Riemannian geometry concepts that we use in this work. The interested reader can refer to [Pet06; Bac14] for an in-depth review of this topic, but for this work the following

notions will be enough. A Riemannian manifold $(\mathcal{M}, \mathfrak{g})$ is a real $C^\infty$ manifold $\mathcal{M}$ equipped with a metric $\mathfrak{g}$, which is a smoothly varying, i.e., $C^\infty$, inner product. For $x \in \mathcal{M}$, denote by $T_x\mathcal{M}$ the tangent space of $\mathcal{M}$ at $x$. For vectors $v, w \in T_x\mathcal{M}$, we denote the inner product of the metric by $\langle v, w \rangle_x$ and the norm it induces by $\|v\|_x \overset{\text{def}}{=} \sqrt{\langle v, v \rangle_x}$. Most of the time, the point $x$ is known from context, in which case we write $\langle v, w \rangle$ or $\|v\|$.

A geodesic of length $\ell$ is a curve $\gamma : [0, \ell] \to \mathcal{M}$ of unit speed that is locally distance minimizing. A uniquely geodesic space is a space such that for every two points there is one and only one geodesic that joins them. In such a case the exponential map $\mathrm{Exp}_x : T_x\mathcal{M} \to \mathcal{M}$ and the inverse exponential map $\mathrm{Log}_x : \mathcal{M} \to T_x\mathcal{M}$ are well defined for every pair of points, and are as follows. Given $x, y \in \mathcal{M}$, $v \in T_x\mathcal{M}$, and a geodesic $\gamma$ of length $\|v\|$ such that $\gamma(0) = x$, $\gamma(\|v\|) = y$, $\gamma'(0) = v/\|v\|$, we have that $\mathrm{Exp}_x(v) = y$ and $\mathrm{Log}_x(y) = v$. We denote by $d(x, y)$ the distance between $x$ and $y$, and note that it takes the same value as $\|\mathrm{Log}_x(y)\|$. The manifold $\mathcal{M}$ comes with a natural parallel transport between vectors in different tangent spaces, that formally is defined from a way of identifying nearby tangent spaces, known as the Levi-Civita connection $\nabla$ [Lev77]. We use this parallel transport throughout this work.

Given a 2-dimensional subspace $V \subseteq T_x\mathcal{M}$ of the tangent space of a point $x$, the sectional curvature at $x$ with respect to $V$ is defined as the Gauss curvature, for the surface $\mathrm{Exp}_x(V)$ at $x$. The Gauss curvature at a point $x$ can be defined as the product of the maximum and minimum curvatures of the curves resulting from intersecting the surface with planes that are normal to the surface at $x$. A Hadamard manifold is a complete simply connected Riemannian manifold whose sectional curvature is non-positive, like the hyperbolic space or the space of $n \times n$ symmetric positive definite matrices with the metric $\langle X, Y \rangle_A \overset{\text{def}}{=} \mathrm{Tr}(A^{-1}XA^{-1}Y)$ where $X, Y$ are in the tangent space of $A$. Hadamard manifolds are uniquely geodesic. Note that in a general manifold $\mathrm{Exp}_x(\cdot)$ might not be defined for each $v \in T_x\mathcal{M}$, but in a Hadamard manifold of dimension $n$, the exponential map at any point is a global diffeomorphism between $T_x\mathcal{M} \cong \mathbb{R}^n$ and the manifold, and so the exponential map is defined everywhere. We now proceed to define the main properties that will be assumed on our model for the function to be minimized and on the feasible set $\mathcal{X}$.

**Definition 1.1 (Geodesic Convexity and Smoothness).** Let $f : \mathcal{N} \subset \mathcal{M} \to \mathbb{R}$ be a differentiable function defined on an open set $\mathcal{N}$ contained in a Riemannian manifold $\mathcal{M}$. Given $L \geq \mu > 0$, we say that $f$ is $L$-*smooth* in $\mathcal{X}$ if for any two points $x, y \in \mathcal{X}$, $f$ satisfies

$$f(y) \leq f(x) + \langle \nabla f(x), \mathrm{Log}_x(y) \rangle + \frac{L}{2}d(x, y)^2.$$

Analogously, we say that $f$ is $\mu$-*strongly g-convex* in $\mathcal{X}$, if for any two points $x, y \in \mathcal{X}$, we have

$$f(y) \geq f(x) + \langle \nabla f(x), \mathrm{Log}_x(y) \rangle + \frac{\mu}{2}d(x, y)^2.$$

If the previous inequality is satisfied with $\mu = 0$, we say the function is *g-convex* in $\mathcal{X}$.

**Definition 1.2 (Metric projection operator).** Let $\mathcal{M}$ be a Hadamard manifold and let $\mathcal{X} \subset \mathcal{M}$ be a closed g-convex subset of $\mathcal{M}$. A *metric projection operator* onto $\mathcal{X}$ is a map $\mathcal{P}_\mathcal{X} : \mathcal{M} \to \mathcal{X}$ satisfying $d(x, \mathcal{P}_\mathcal{X}(x)) \leq d(x, y)$ for all $y \in \mathcal{X}$.

A consequence of the definition is that the projection is single valued and non-expansive, the latter meaning $d(\mathcal{P}_\mathcal{X}(x), \mathcal{P}_\mathcal{X}(y)) \leq d(x, y)$, cf. [Bac14, Thm 2.1.12].

We present the following fact about the squared distance function, when one of the arguments is fixed. The constants $\zeta_D$, $\delta_D$ below appear everywhere in Riemannian optimization because, among other things, Fact 1.3 yields Riemannian inequalities that are analogous to the equality in the Euclidean cosine law of a triangle, cf. Corollary B.3, and these inequalities have wide applicability in the analyses of Riemannian methods.

**Fact 1.3 (Local information of the squared distance).** *Let $\mathcal{M}$ be a Riemannian manifold of sectional curvature bounded by $[\kappa_{\min}, \kappa_{\max}]$ that contains a uniquely g-convex set $\mathcal{X} \subset \mathcal{M}$ of diameter $D < \infty$. Then, given $x, y \in \mathcal{X}$ we have the following for the function $\Phi_x : \mathcal{M} \to \mathbb{R}$, $y \mapsto \frac{1}{2}d(x, y)^2$:*

$$\nabla\Phi_x(y) = -\mathrm{Log}_y(x) \qquad \text{and} \qquad \delta_D\|v\|^2 \leq \mathrm{Hess}\,\Phi_x(y)[v, v] \leq \zeta_D\|v\|^2,$$

*where*

$$\zeta_D \overset{\text{def}}{=} \begin{cases} D\sqrt{|\kappa_{\min}|}\coth(D\sqrt{|\kappa_{\min}|}) & \text{if } \kappa_{\min} \leq 0 \\ 1 & \text{if } \kappa_{\min} > 0 \end{cases},$$

133 *and*

$$\delta_D \stackrel{\text{def}}{=} \begin{cases} 1 & \text{if } \kappa_{\max} \leq 0 \\ D\sqrt{\kappa_{\max}} \cot(D\sqrt{\kappa_{\max}}) & \text{if } \kappa_{\max} > 0 \end{cases},$$

134 *In particular, $\Phi_x$ is $\delta_D$-strongly g-convex and $\zeta_D$-smooth in $\mathcal{X}$. See [Lez20] for a proof.*

## 1.2 Notation.

136 Let $\mathcal{M}$ be a uniquely geodesic $n$-dimensional Riemannian manifold. Given points $x, y, z \in \mathcal{M}$,
137 we abuse the notation and write $y$ in non-ambiguous and well-defined contexts in which we should
138 write $\text{Log}_x(y)$. For example, for $v \in T_x\mathcal{M}$ we have $\langle v, y - x \rangle = -\langle v, x - y \rangle = \langle v, \text{Log}_x(y) -$
139 $\text{Log}_x(x) \rangle = \langle v, \text{Log}_x(y) \rangle$; $\|v - y\| = \|v - \text{Log}_x(y)\|$; $\|z - y\|_x = \|\text{Log}_x(z) - \text{Log}_x(y)\|$; and
140 $\|y - x\|_x = \|\text{Log}_x(y)\| = d(y, x)$. We denote by $\mathcal{X}$ a compact, uniquely geodesic g-convex set of
141 diameter $D$ contained in an open set $\mathcal{N} \subset \mathcal{M}$ and we use $I_{\mathcal{X}}$ for the indicator function of $\mathcal{X}$, which
142 is 0 at points in $\mathcal{X}$ and $+\infty$ otherwise. For a vector $v \in T_y\mathcal{M}$, we use $\Gamma_y^x(v) \in T_x\mathcal{M}$ to denote the
143 parallel transport of $v$ from $T_y\mathcal{M}$ to $T_x\mathcal{M}$ along the unique geodesic that connects $y$ to $x$. We call
144 $f : \mathcal{N} \subset \mathcal{M} \to \mathbb{R}$ a differentiable $L$-smooth g-convex function we want to optimize over $\mathcal{X}$. We use
145 $\varepsilon$ to denote the approximation accuracy parameter, $x_0 \in \mathcal{X}$ for the initial point of our algorithms, and
146 $R_0 \stackrel{\text{def}}{=} d(x_0, x^*)$ for the initial distance to an arbitrary minimizer $x^* \in \arg\min_{x \in \mathcal{X}} f(x)$. The big
147 $O$ notation $\widetilde{O}(\cdot)$ omits log factors and $O^*(\cdot)$ omits log factors except those with respect to $LR_0^2/\varepsilon$.
148 The latter will be useful to describe the rates of convergence for the strongly g-convex case, by
149 emphasizing that there is no extra dependence on $\varepsilon$. Note that in the setting of Hadamard manifolds,
150 the bounds on the sectional curvature are $\kappa_{\min} \leq \kappa_{\max} \leq 0$. Hence for convenience, given that we
151 optimize over $\mathcal{X}$, we define $\zeta \stackrel{\text{def}}{=} \zeta_D = D\sqrt{|\kappa_{\min}|} \coth(D\sqrt{|\kappa_{\min}|}) \geq 1$ and $\delta \stackrel{\text{def}}{=} 1$. If $v \in T_x\mathcal{M}$,
152 we use $\Pi_{\bar{B}(0,D)}(v) \in T_x\mathcal{M}$ for the projection of $v$ onto the closed ball with center at 0 and radius $D$.

## 1.3 Our results and comparisons with related work

154 In this work, we optimize functions defined over Hadamard manifolds $\mathcal{M}$ of finite dimension $n$
155 and of sectional curvature bounded in $[\kappa_{\min}, \kappa_{\max}]$. As all previous related works discussed in the
156 sequel, we assume that we can compute the exponential and inverse exponential maps, and parallel
157 transport of vectors for our manifold. The differentiable function $f$ to be optimized is defined over
158 an open set $\mathcal{N} \subset \mathcal{M}$ that contains a compact g-convex set $\mathcal{X}$ of finite diameter $D$, that we access
159 via a metric-projection oracle. Our function $f$ is $L$-smooth and g-convex (or $\mu$-strongly g-convex)
160 in $\mathcal{X}$ and we have access to it via a gradient oracle that can be queried at points in $\mathcal{X}$. For the
161 setting we just described, we show in Theorem 2.2 and Theorem 2.4 that the algorithms we propose
162 find a point $y_T \in \mathcal{X}$ such that $f(y_T) - \min_{x \in \mathcal{X}} f(x) \leq \varepsilon$ after calling the gradient oracle and the
163 metric-projection oracle the following number of times: $\widetilde{O}(\zeta^2\sqrt{LR_0^2/\varepsilon})$ for the g-convex case and
164 $O^*(\zeta^2\sqrt{L/\mu}\log(\mu R_0^2/\varepsilon))$ for the $\mu$-strongly g-convex case, where $R_0 \stackrel{\text{def}}{=} d(x_0, x^*)$ and $x_0 \in \mathcal{X}$ is
165 an initial point. That is, the algorithms enjoy the same rates as AGD in the Euclidean space up to a
166 factor of $\zeta^2 = D^2\kappa_{\min}^2 \coth^2(D\sqrt{|\kappa_{\min}|})$ (our geometric penalty) and up to universal constants and
167 log factors. Note that as the minimum curvature $\kappa_{\min}$ approaches 0 we have $\zeta \to 1$.

168 We emphasize that our algorithms only need to query the gradient of $f$ at points in $\mathcal{X}$ and the
169 $L$-smoothness and $\mu$-strong g-convexity of $f$ only need to hold in $\mathcal{X}$. This is relevant because in
170 Riemannian manifolds the condition number $L/\mu$ in a set can increase with the size of the set, cf.
171 [Mar22, Proposition 27]. Intuitively, although there are twice differentiable functions defined over the
172 Euclidean space whose Hessian is constant everywhere, in other Riemannian cases the metric may
173 preclude having such global condition and the larger the (compact) set is, the greater the maximum
174 eigenvalue of the Hessian over this set (i.e., its smoothness constant) can be *for any smooth and*
175 *strongly g-convex function*. And similarly with the minimum one, i.e., its strong g-convexity constant.
176 Compare this, for instance, with the bounds on the Hessian's eigenvalues of the squared distance
177 function in Fact 1.3, which are tight for spaces of constant curvature [Lez20].

178 Now we proceed to compare our results with previous works. We have summarized most of the
179 following discussion in Table 1. We include Nesterov's AGD in the table for comparison purposes[1].

---

[1] Note that the original method in [Nes83] needed to query the gradient of the function outside of the feasible set, and this was later improved to only require queries at feasible points [Nes05] as in our work, hence our choice of citation in the table.

There are some works on Riemannian acceleration that focus on empirical evaluation or that work under strong assumptions [Liu+17; Ali+19; HW19a; Ali+20; Lin+20], see [Mar22] for instance for a discussion on these works. We focus the discussion on the most related work with guarantees. [ZS18] obtain an algorithm that, up to constants, achieves the same rates as AGD in the Euclidean space, for $L$-smooth and $\mu$-strongly g-convex functions but only *locally*, namely when the initial point starts in a small neighborhood $N$ of the minimizer $x^*$: a ball of radius $O((\mu/L)^{3/4})$ around it. [AS20] generalize the previous algorithm and, by using similar ideas as in [ZS18] for estimating a lower bound on $f$, they adapt the algorithm to work globally, proving that it eventually decreases the objective as fast as AGD. However, as [Mar22] noted, it takes as many iterations as the ones needed by RGD to reach the neighborhood of the previous algorithm. The latter work also noted that in fact RGD and the algorithm in [ZS18] can be run in parallel and combined to obtain the same convergence rates as in [AS20], which suggested that for this technique, full acceleration with the rates of AGD only happens over the small neighborhood $N$ in [ZS18]. Note however that [AS20] show that their algorithm will decrease the function value faster than RGD, but this is not quantified. [JS21] developed a different framework, arising from [AS20] but with the same guarantees for accelerated first-order methods. We do not feature it in the table. [CB21] showed that in a ball of center $x \in \mathcal{M}$ and radius $O((\mu/L)^{1/2})$ containing $x^*$, the pullback function $f \circ \mathrm{Exp}_x : T_x\mathcal{M} \to \mathbb{R}$ is strongly convex and smooth with condition number $O(L/\mu)$, so they argue that using AGD on the pullback over the corresponding pulled-back Euclidean ball in the tangent space results in local acceleration as well. In short, acceleration is possible in a small neighborhood because there the manifold is almost Euclidean and the geometric deformations are small in comparison to the curvature of the objective. These techniques do not work with the g-convex case since the neighborhood becomes a point ($\mu/L = 0$).

Finding fully accelerated algorithms that are *global* presents a harder challenge. By a fully accelerated algorithm we mean one with rates with same dependence as AGD on $L$, $\varepsilon$, and if it applies, on $\mu$. [Mar22] provided such algorithms for g-convex functions, strongly or not, defined over manifolds of constant sectional curvature and constrained to a ball of radius $R$. In the convergence rates, there is a geometric factor of $c = \cos(R\sqrt{K})^{-\Theta(1)}$ for sectional curvature $K > 0$, and $c = \cosh(R\sqrt{-K})^{\Theta(1)}$ when $K < 0$, cf. Table 1. When $R\sqrt{|K|} = O(1)$, they recover the same rates as AGD, which for those manifolds is more general than the local assumption in the previous set of works. For larger values of $R\sqrt{|K|}$, there is also full acceleration, but note that $c$ grows rapidly when $K < 0$, since there is an exponential dependence on $R$. When $K > 0$ the geometric penalty also grows fast, but this is more natural since the minimum condition number of a function in a ball of radius $R$ grows similarly [Mar22]. The geometric penalties are large in some regimes because the algorithm bounds uniformly, over the whole domain, the worst-case deformations that can occur. On the other hand, for manifolds of bounded sectional curvature, [KY22] design algorithms with the same rates as AGD up to universal constants and a factor of $\zeta$, their geometric penalty . However, they need to assume that the iterates of their algorithm remain in $\mathcal{X}$ and point out on the necessity of removing such an assumption, which they leave as an open question. Our work solves this question for the case of Hadamard manifolds. In their technique, they show that they can use the structure of the accelerated scheme to *move* lower bound estimations on $f(x^*)$ from one particular tangent space to another without incurring extra errors, when the right Lyapunov function is used. By *moving* lower bounds here we mean finding suitable lower bounds that are simple (a quadratic in their case), if pulled-back to one tangent space, if we start with a similar bound that is simple when pulled-back to another tangent space.

**Lower bounds.** In this paragraph, we omit constants depending on the curvature bounds in the big-$O$ notations for simplicity. [HM21] proved an optimization lower bound showing that acceleration in Riemannian manifolds is harder than in the Euclidean space. [CB21] largely generalized their results. They essentially show that for a large family of Hadamard manifolds, there is a function that is smooth and strongly g-convex in a ball of radius $R$ that contains the minimizer $x^*$, and for which finding a point that is $R/5$ close to $x^*$ requires $\widetilde{\Omega}(R)$ calls to the gradient oracle. Note that these results do not preclude the existence of a fully accelerated algorithm with rates $\widetilde{O}(R)$+AGD rates, for instance. But they show that even if we want to perform unconstrained optimization, so no in-manifold constraints are originally imposed, we need to optimize over a bounded domain in order to bound geometric penalties. A similar statement is provided in the case of smooth and only g-convex functions.

Table 1: Convergence rates of related works with provable guarantees for smooth problems over uniquely geodesic manifolds, in chronological order with respect to when the works were publicly available. Column **K?** refers to the supported values of the sectional curvature, **G?** to whether the algorithm is global (any initial distance to a minimizer is allowed). Here L and L$'$ mean they are local algorithms that require initial distance $O((L/\mu)^{-3/4})$ and $O((L/\mu)^{-1/2})$, respectively. Column **F?** refers to whether there is full acceleration, meaning dependence on $L$, $\mu$, and $\varepsilon$ like AGD up to possibly log factors. Column **C?** refers to whether the method supports constraints. All methods require their iterates to be in some specified compact set, but the works with ✗ just assume the iterates will remain within the constraints, while the ones with ✓ can force this condition with a projection oracle. Also, here B is like ✓ but with the constraints limited to a ball. See Section 1.3 for the value $c$ in [Mar22]. We use $\mathcal{W} \stackrel{\text{def}}{=} \sqrt{\frac{L}{\mu}} \log(\frac{LR_0^2}{\varepsilon})$. $^*$In [CB21], a condition is required on the covariant derivative of the metric tensor, cf. [CB21, Section 6].

| Method | g-convex | $\mu$-st. g-convex | K? | G? | F? | C? |
|---|---|---|---|---|---|---|
| [Nes05, AGD] | $O(\sqrt{\frac{LR_0^2}{\varepsilon}})$ | $O(\mathcal{W})$ | 0 | ✓ | ✓ | ✓ |
| [ZS18, Theorem 11] | - | $O(\mathcal{W})$ | bounded | L | ✓ | ✗ |
| [AS20, Theorem 3.1] | - | $O^*(\frac{L}{\mu} + \mathcal{W})$ | bounded | ✓ | ✗ | ✗ |
| [Mar22, Remark 30] | - | $O^*(\frac{L}{\mu} + \mathcal{W})$ | bounded | ✓ | ✗ | ✗ |
| [Mar22, Theorems 6 & 8] | $\widetilde{O}(c\sqrt{\frac{LR_0^2}{\varepsilon}})$ | $O^*(c \cdot \mathcal{W})$ | ctant.$\neq 0$ | ✓ | ✓ | B |
| [CB21, Section 6] | - | $O(\mathcal{W})$ | bounded$^*$ | L$'$ | ✓ | B |
| [KY22, Corollaries 1 & 2] | $O(\zeta\sqrt{\frac{LR_0^2}{\varepsilon}})$ | $O(\zeta \cdot \mathcal{W})$ | bounded | ✓ | ✓ | ✗ |
| **Theorems 2.2 & 2.4** | $\widetilde{O}(\zeta^2\sqrt{\frac{LR_0^2}{\varepsilon}})$ | $O^*(\zeta^2 \cdot \mathcal{W})$ | Hadamard | ✓ | ✓ | ✓ |

**Handling constraints to bound geometric penalties.** Due to the lower bounds, it becomes crucial for a fully accelerated algorithm to restrict the optimization to a set $\mathcal{X}$ of finite diameter $D$, or otherwise a worst-case analysis incurs an arbitrary large geometric penalty in the rates. In our algorithm and in all other known fully accelerated algorithms, learning rates depend on this diameter. This is natural: estimation errors due to geometric deformations depend on the diameter via the constants $\zeta_D$, $\delta_D$, the cosine-law inequalities Corollary B.3, or other analogous inequalities, and the algorithms take these errors into account. All other previous works are not able to deal with any constraints and hence they simply assume that the iterates of their algorithms stay within one such specified set, except for [Mar22] and [CB21] that enforce a ball constraint, as we explained above. However, these two works have their applicability limited to spaces of constant curvature and to local optimization, respectively. Note that even if one could show in some settings that given a choice of learning rate, convergence implies that the iterates will remain in some compact set, then because the learning rates depend on the diameter of the set, and the diameter of the set would depend on the learning rates, one cannot conclude from this argument that the assumption these works make is going to be satisfied. In contrast, in this work, we design the first accelerated algorithm that supports metric projections and, consequently, we can handle general constraints to bound geometric penalties and accelerate our method without any other extra assumptions, solving an open question in [KY22].

Some other works study and use Riemannian metric projections in other contexts, see [Wal74; HP13; BHP13; Bac14; ZS16] and references therein. Among them, [ZS16] introduced several, both deterministic and stochastic, *unaccelerated* first-order methods that work with in-manifold constraints by using metric-projection oracles. Our Algorithm 1 uses their projected RGD as a subroutine, cf. Remark 2.3.

**Finding a global minimizer.** In our work, we do not need to assume that the set $\mathcal{X}$ contains a global minimizer, namely a point $x^*$ such that $\nabla f(x^*) = 0$. We find an $\varepsilon$-minimizer with respect to the minimum value of $f$ at $\mathcal{X}$. All other previous works assume that the set contains the minimizers of $f$, with the exception of [Mar22], where the algorithm can forgo this assumption if one has access to a bound $L_{f,\mathcal{B}}$ on the Lipschitz constant of $f$ when restricted to their ball constraint $\mathcal{B}$, and in such a case the rates have a $\log(L_{f,\mathcal{B}} D/\varepsilon)$ factor instead of a $\log(LD^2/\varepsilon)$ factor. Note this is natural since if a global minimizer is in the set, then we have $L_{f,\mathcal{B}} = O(LD)$. We note that we also

obtain a logarithmic dependence that involves the Lipschitz constant $L_{f,\mathcal{X}}$ of $f$ in $\mathcal{X}$ (the logarithmic dependence involves the scale invariant quantity $\zeta_C$ for $C = L_{f,\mathcal{X}}/L$, which is $O(\zeta)$ if $x^* \in \mathcal{X}$) but in contrast in our case, our method does not require access to the Lipschitz constant of $f$ in $\mathcal{X}$.

**Riemannian proximal methods** There have been some works that study proximal methods in Riemannian manifolds, but most of them focus on asymptotic results or assume the proximal operator can be computed exactly [Wan+15; BFM17; BCO16; Kha+21; Cha+21]. The rest of these works study proximal point methods under different inexact versions of the proximal operator as ours and they do not show how to implement their inexact version in applications, like our case of smooth and g-convex optimization. [AK14] provide a convergence analysis of an inexact proximal point method but when applied to optimization they assume the computation of the proximal operator is exact. [TH14] uses a different inexact condition and proves linear convergence, under a growth condition on $f$. [Wan+16] obtains linear convergence of an inexact proximal point method under a different growth assumption on $f$ and under an absolute error condition on the proximal function.

# 2 Algorithm and Pseudocode

In this section, we present our **Riema**nnian **ac**celerated algorithm for **con**strained g-convex optimization, or Riemacon[2]. Recall our abuse of notation for points $p \in \mathcal{M}$ to mean $\mathrm{Log}_q(p)$ in contexts in which one should place a vector in $T_q\mathcal{M}$ and note that in our algorithm $x_k$ and $y_k$ are points in $\mathcal{M}$ whereas $z_k^{x_k} \in T_{x_k}\mathcal{M}, z_k^{y_k}, \bar{z}_k^{y_k} \in T_{y_k}\mathcal{M}$.

---

**Algorithm 1** Riemacon: **Riema**nnian **Ac**celeration - **Con**strained g-Convex Optimization

---

**Input:** Initial point $x_0 \in \mathcal{X} \subset \mathcal{N}$. Diff. function $f : \mathcal{N} \subset \mathcal{M} \to \mathbb{R}$ for a Hadamard manifold $\mathcal{M}$ that is $L$-smooth and g-convex in $\mathcal{X}$, final iteration $T$ (not required to be known in advance).

    **Parameters:**

- Geometric penalty $\xi \stackrel{\mathrm{def}}{=} 4\zeta_{2D} - 3 \le 8\zeta - 3 = O(\zeta)$.
- Implicit Gradient Descent learning rate $\lambda \stackrel{\mathrm{def}}{=} \zeta_{2D}/L$.
- Mirror Descent learning rates $\eta_k \stackrel{\mathrm{def}}{=} a_k/\xi$.
- Proportionality constant in the proximal subproblem accuracies: $\Delta_k \stackrel{\mathrm{def}}{=} \frac{1}{(k+1)^2}$.

    **Definition:** (computation of this value is not needed)

- Prox. accuracies: $\sigma_k \stackrel{\mathrm{def}}{=} \frac{\Delta_k d(x_k, y_k^*)^2}{78\lambda}$ where $y_k^* \stackrel{\mathrm{def}}{=} \arg\min_{y \in \mathcal{X}}\{f(y) + \frac{1}{2\lambda}d(x_k, y)^2\}$.

---

1:   $y_0 \leftarrow x_0;$     $A_0 \leftarrow 200\lambda\xi$
2:   $z_0^{x_0} \leftarrow 0 \in T_{x_0}\mathcal{M};$    $\bar{z}_0^{y_0} \leftarrow z_0^{y_0} \leftarrow 0 \in T_{y_0}\mathcal{M}$
3:   **for** $k = 1$ **to** $T$ **do**
4:      $a_k \leftarrow 2\lambda\frac{k+32\xi}{5}$
5:      $A_k \leftarrow a_k/\xi + A_{k-1} = \sum_{i=1}^k a_i/\xi + A_0 = \lambda\left(\frac{k(k+1+64\xi)}{5\xi} + 200\xi\right)$
6:      $x_k \leftarrow \mathrm{Exp}_{y_{k-1}}\left(\frac{a_k}{A_{k-1}+a_k}\bar{z}_{k-1}^{y_{k-1}} + \frac{A_{k-1}}{A_{k-1}+a_k}y_{k-1}\right) = \mathrm{Exp}_{y_{k-1}}\left(\frac{a_k}{A_{k-1}+a_k}\bar{z}_{k-1}^{y_{k-1}}\right)$    $\diamond$ Coupling
7:      $z_{k-1}^{x_k} \leftarrow \Gamma_{y_{k-1}}^{x_k}(\bar{z}_{k-1}^{y_{k-1}}) + \mathrm{Log}_{x_k}(y_{k-1}) = \mathrm{Log}_{x_k}(\mathrm{Exp}_{y_k}(\bar{z}_{k-1}^{y_{k-1}}))$
8:      $y_k \leftarrow \sigma_k$-minimizer of the proximal problem $\min_{y \in \mathcal{X}}\{f(y) + \frac{1}{2\lambda}d(x_k, y)^2\}$ (cf. Remark 2.3).
9:      $v_k^x \leftarrow -\mathrm{Log}_{x_k}(y_k)/\lambda$                               $\diamond$ Approximate subgradient
10:     $z_k^{x_k} \leftarrow z_{k-1}^{x_k} - \eta_k v_k^x$                                   $\diamond$ Mirror Descent step
11:     $z_k^{y_k} \leftarrow \Gamma_{x_k}^{y_k}(z_k^{x_k}) + \mathrm{Log}_{y_k}(x_k)$                $\diamond$ Moving the dual point to $T_{y_k}\mathcal{M}$
12:     $\bar{z}_k^{y_k} \leftarrow \Pi_{\bar{B}(0,D)}(z_k^{y_k}) \in T_{y_k}\mathcal{M}$      $\diamond$ Easy projection done so the dual point is not very far
13: **end for**
14: **return** $y_T$.

---

We start with an interpretation of our algorithm that helps understanding its high-level ideas. The following intends to be a qualitative explanation, and we refer to the pseudocode and the supplementary material for the exact descriptions and analysis. Euclidean accelerated algorithms can be interpreted, cf. [AO17], as a combination of a gradient descent (GD) algorithm and an online learning algorithm

---

[2]Riemacon rhymes with "rima con" in Spanish.

with losses being the affine lower bounds $f(x_k) + \langle \nabla f(x_k), \cdot - x_k \rangle$ we obtain on $f(\cdot)$ by applying convexity at some points $x_k$. That is, the latter builds a lower bound estimation on $f$. By selecting the next query to the gradient oracle as a cleverly picked convex combination of the predictions given by these two algorithms, one can show that the instantaneous regret of the online learning algorithm can be compensated by the local progress GD makes, which leads to accelerated convergence. In Riemannian optimization, there are two main obstacles. Firstly, the first-order approximation of $f$ at a point $x$ yields functions that are affine but only with respect to its respective $T_x\mathcal{M}$, and so combining these lower bounds that are only simple in their tangent spaces makes obtaining good global estimations not simple. Secondly, when one obtains such global estimations, then one naturally incurs an instantaneous regret that is worse by a factor than is usual in Euclidean acceleration. This factor is a geometric constant depending on the diameter $D$ of a set $\mathcal{X}$ where the iterates and a (possibly constrained) minimizer lie. As a consequence, the learning rate of GD would need to be multiplicatively increased by such a constant with respect to the one of the online learning algorithm in order for the regret to still be compensated with the local progress of GD (and the rates worsen by this constant). But if we fix some $\mathcal{X}$ of finite diameter, because GD's learning rate is now larger, it is not clear how to keep the iterates in $\mathcal{X}$. And if we do not have the iterates in one such set $\mathcal{X}$, then our geometric penalties could grow arbitrarily.

We find the answer in implicit methods. An implicit Euclidean (sub)gradient descent step is one that computes, from a point $x_k \in \mathcal{X}$, another point $y_k^* = x_k - \lambda v_k \in \mathcal{X}$, where $v_k \in \partial(f + I_\mathcal{X})(y_k^*)$, is a subgradient of $f + I_\mathcal{X}$ at $y_k^*$. Intuitively, if we could implement a Riemannian version of an implicit GD step then it should be possible to still compensate the regret of the other algorithm and keep all the iterates in the set $\mathcal{X}$. Computing such an implicit step is computationally hard in general, but we show that approximating the proximal objective $h_k(y) \stackrel{\text{def}}{=} f(y) + \frac{1}{2\lambda}d(x_k, y)^2$ with enough accuracy yields an approximate subgradient that can be used to obtain an accelerated algorithm as well. In particular, we provide an accelerated scheme for which we show that the error incurred by the approximation of the subgradient can be bounded by some terms we can control, cf. Lemma A.2, namely a small term that appears in our Lyapunov function and also a term proportional to the squared norm of the approximated subgradient, which only adds a constant to the final convergence rates. We also provide a warm start in Lemma A.4 and an analysis that shows that using the projected Riemannian gradient descent in [ZS18] initialized at the warm start point achieves the desired accuracy of the subproblem fast, cf. Remark 2.3. This proximal approach works by exploiting the fact that the Riemannian Moreau envelop is convex in Hadamard manifolds [AF05] and that the subproblem $h_k$, defined with our $\lambda = \zeta_{2D}/L$, is strongly g-convex and smooth with a condition number that only depends on the geometry. Besides of these steps, we use a coupling of the approximate implicit RGD and of a mirror descent (MD) algorithm, along with a technique in [KY22] to move dual points to the right tangent spaces without incurring extra geometric penalties, that we adapt to work with dual projections, cf. Lemma A.3. Importantly, the MD algorithm keeps the dual point close to the set $\mathcal{X}$ by using the projection in Line 12, which implies that the point $x_k$ is close to $\mathcal{X}$ as well, and this is crucial to keep low geometric penalties. This MD approach is a mix between follow-the-regularized-leader algorithms, that do not project the dual variable, and pure mirror descent algorithms that always project the dual variable. In the analysis, we note that partial projection also works, meaning that defining a new dual point that is closer to all of the points in the feasible set but without being a full projection leads to the same guarantees. Because we use the mirror descent lemma over $T_{y_k}\mathcal{M}$, what we described translates to: we can project the dual $z_k^{y_k}$ onto a ball defined on $T_{y_k}\mathcal{M}$ that contains the pulled-back set $\text{Log}_{y_k}(\mathcal{X})$ and by means of that trick we can keep the iterates $x_k$ close to $\mathcal{X}$. And at the same time, the point for which we prove guarantees, namely $y_k$, is always in $\mathcal{X}$.

We leave the proofs of most of our results to the supplementary material and state our main theorems below. Using the insights explained above, we show the following inequality on $\psi_k$, defined below, that will be used as a Lypapunov function to prove the convergence rates of Algorithm 1.

**Proposition 2.1.** [↓] *By using the notation of Algorithm 1, let*

$$\psi_k \stackrel{\text{def}}{=} A_k(f(y_k) - f(x^*)) + \frac{1}{2}\|z_k^{y_k} - x^*\|_{y_k}^2 + \frac{\xi - 1}{2}\|y_k - z_k^{y_k}\|_{y_k}^2.$$

*Then, for all $k \geq 1$, we have $(1 - \Delta_k)\psi_k \leq \psi_{k-1}$.*

Finally, we can state our theorem for the optimization of $L$-smooth and g-convex functions.

**Theorem 2.2.** [↓] *Let $\mathcal{M}$ be a finite-dimensional Hadamard manifold of bounded sectional curvature, let $f : \mathcal{N} \subset \mathcal{M} \to \mathbb{R}$ be an $L$-smooth and g-convex differentiable function in a compact g-convex*

set $\mathcal{X} \subset \mathcal{N}$ of diameter $D$, and $x^* \in \arg\min_{x \in \mathcal{X}} f(x)$. For $R_0 \overset{\text{def}}{=} d(x_0, x^*)$, and all $k \geq 1$, the iterates $y_k$ of Algorithm 1 satisfy $y_k \in \mathcal{X}$ and $f(y_k) - f(x^*) = O\left(\frac{LR_0^2}{k^2} \cdot \zeta^2\right)$. That is, after $T = O(\zeta\sqrt{\frac{LR_0^2}{\varepsilon}})$ iterations we find an $\varepsilon$-minimizer. Moreover, the total number of queries to the gradient and projection oracles can be bounded by $\widetilde{O}(\zeta^2\sqrt{\frac{LR_0^2}{\varepsilon}})$.

We note that a straightforward corollary from our results is that if we can compute the exact Riemannian proximal point operator and we use it as the implicit gradient descent step in Line 8 of Algorithm 1, then the method is an accelerated proximal point method. One such Riemannian algorithm was previously unknown in the literature as well.

Now we show that Line 8 can be implemented efficiently. The essential part is being able to have and use a point with the guarantees of our warm start, cf. Lemma A.4.

**Remark 2.3 (Solving the subproblems).** *Let $\mathcal{A}$ be the unaccelerated Riemannian gradient descent algorithm in [ZS16, Theorem 15]. This algorithm takes a function $h : \mathcal{M} \to \mathbb{R}$ with minimizer at $y^*$ when restricted to $\mathcal{X} \subset \mathcal{M}$ that is $\mu'$-strongly g-convex and $L'$-smooth in $\mathcal{X}$, where $\mathcal{M}$ is a Hadamard manifold of bounded sectional curvature and $\mathcal{X}$ is a geodesically-convex compact set with diameter $D$ and returns a point $p_t$ satisfying $h_k(p_t) - h_k(y^*) \leq \varepsilon'$ after querying a gradient oracle for $h_k$ and a metric-projection oracle $\mathcal{P}_\mathcal{X}$ for $\mathcal{X}$ for $t = O((\zeta + \frac{L'}{\mu'})\log(\frac{(h_k(p_0) - h_k(y^*)) + L'd(p_0, y^*)^2}{\varepsilon'}))$ times[3]. If we apply this algorithm to $h \leftarrow h_k(y) \overset{\text{def}}{=} f(y) + \frac{1}{2\lambda}d(x_k, y)^2$, we have $y^* \leftarrow y_k^*$, $L' \leftarrow 2L$ and $\mu' \leftarrow L/\zeta_{2D}$, so the condition number is $L'/\mu' = O(\zeta_{2D}) = O(\zeta)$. This is computed taking into account that $f$ is $L$-smooth and $0$-strongly g-convex and using the $\zeta_{2D}/\lambda$-smoothness and $1/\lambda$-strong g-convexity of the second summand, which is given by Fact 1.3 and (1). If we initialize the method with $p_0 \overset{\text{def}}{=} \mathcal{P}_\mathcal{X}(\text{Exp}_{x_k'}(-\frac{1}{L'}\nabla h_k(x_k')))$, where $x_k' \overset{\text{def}}{=} \mathcal{P}_\mathcal{X}(x_k)$, then using $(L/\zeta_{2D})$-strong g-convexity of $h_k$ to bound $L'd(p_0, y_k^*)^2 \leq 4\zeta_{2D}(h_k(p_0) - h(y_k^*))$, using Lemma A.4 with $x \leftarrow x_k$, $p \leftarrow y_k^*$, and using the guarantees on $\mathcal{A}$, we have that we find a point $y_k$ satisfying $h_k(y_k) - h_k(y_k^*) \leq \frac{\Delta_k d(x_k, y_k^*)^2}{78\lambda}$ in $\widetilde{O}(\zeta)$ queries to the gradient and projection oracles. See Remark A.5 for the computation of this value. We note that any other algorithm with linear convergence rates for constrained strongly g-convex, smooth problems that works with a metric-projection oracle can be used as a subroutine to obtain an accelerated Riemannian algorithm.*

We introduce the algorithm for $\mu$-strongly g-convex functions via a reduction to Algorithm 1, for simplicity. We note that the reverse Riemannian reduction yields extra factors in the rates depending on $R_0$ and the curvature, but this reduction does not yield any extra factors in the rates and it actually is slightly better than the usual convergence that is obtained when one analyzes these kinds of accelerated algorithms directly, by having a $\mu$ factor instead of $L$ inside of the logarithm.

**Theorem 2.4.** [↓] *Under the same assumptions as in Theorem 2.2, let now $f$ be $\mu$-strongly g-convex. Applying the reduction in [Mar22, Theorem 7], we obtain an algorithm that finds an $\varepsilon$-minimizer of $f$ by querying the gradient oracle and projection oracle $O^*(\zeta^2\sqrt{\frac{L}{\mu}}\log(\frac{\mu R_0^2}{\varepsilon}))$ times.*

## 3 Conclusion and future directions

In this work, we pursued an approach that, by designing inexact Riemannian proximal methods, yielded accelerated optimization algorithms that can work with metric projection oracles. Consequently we were able to work without an undesirable assumption that most previous methods required, whose potential satisfiability is not clear: that the iterates stay in certain specified geodesically-convex set without enforcing them to be in the set. A future direction of research is the study of whether there are algorithms like ours that incur even lower geometric penalties or that do not incur $\log(1/\varepsilon)$ factors. Another interesting direction consists of studying generalizations of our approach to manifolds of non-negative or of bounded sectional cuvature manifolds.

---

[3]In their theorem, the authors only stated that $O((\frac{L'}{\mu'} + \zeta)\log(\frac{L'D^2}{\varepsilon'}))$ queries to the gradient oracle are enough, but in their proof they show this more refined statement, that we use.

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

| 496 | [EAS98] | Alan Edelman, Tomás A. Arias, and Steven Thomas Smith. "The Geometry of Algorithms with Orthogonality Constraints". In: *SIAM J. Matrix Analysis Applications* 20.2 (1998), pp. 303–353 (cit. on p. 1). |

[Gas+19] Alexander Gasnikov, Pavel E. Dvurechensky, Eduard A. Gorbunov, Evgeniya A. Vorontsova, Daniil Selikhanovych, César A. Uribe, Bo Jiang, Haoyue Wang, Shuzhong Zhang, Sébastien Bubeck, Qijia Jiang, Yin Tat Lee, Yuanzhi Li, and Aaron Sidford. "Near Optimal Methods for Minimizing Convex Functions with Lipschitz $p$-th Derivatives". In: *Conference on Learning Theory, COLT 2019, 25-28 June 2019, Phoenix, AZ, USA*. 2019, pp. 1392–1393 (cit. on p. 2).

[GHT15] Matthieu Genicot, Wen Huang, and Nickolay T. Trendafilov. "Weakly Correlated Sparse Components with Nearly Orthonormal Loadings". In: *Geometric Science of Information - Second International Conference, GSI 2015, Palaiseau, France, October 28-30, 2015, Proceedings*. 2015, pp. 484–490 (cit. on p. 1).

[HM21] Linus Hamilton and Ankur Moitra. "A No-go Theorem for Acceleration in the Hyperbolic Plane". In: *arXiv preprint arXiv:2101.05657* (2021) (cit. on p. 5).

[HP13] Seyedehsomayeh Hosseini and Mohamad Pouryayevali. "On the metric projection onto prox-regular subsets of Riemannian manifolds". In: *Proceedings of the American Mathematical Society* 141.1 (2013), pp. 233–244 (cit. on p. 6).

[HS15] Reshad Hosseini and Suvrit Sra. "Matrix Manifold Optimization for Gaussian Mixtures". In: *Advances in Neural Information Processing Systems 28: Annual Conference on Neural Information Processing Systems 2015, December 7-12, 2015, Montreal, Quebec, Canada*. 2015, pp. 910–918 (cit. on p. 1).

[HS17] Reshad Hosseini and Suvrit Sra. "An Alternative to EM for Gaussian Mixture Models: Batch and Stochastic Riemannian Optimization". In: *CoRR* abs/1706.03267 (2017) (cit. on p. 1).

[HS18] Gennadij Heidel and Volker Schulz. "A Riemannian trust-region method for low-rank tensor completion". In: *Numerical Lin. Alg. with Applic.* 25.6 (2018) (cit. on p. 1).

[HS20] Reshad Hosseini and Suvrit Sra. "Recent advances in stochastic Riemannian optimization". In: *Handbook of Variational Methods for Nonlinear Geometric Data* (2020), pp. 527–554 (cit. on p. 2).

[HSS19] Oliver Hinder, Aaron Sidford, and Nimit Sharad Sohoni. "Near-Optimal Methods for Minimizing Star-Convex Functions and Beyond". In: *CoRR* abs/1906.11985 (2019) (cit. on p. 2).

[HW19a] Wen Huang and Ke Wei. "Extending FISTA to Riemannian Optimization for Sparse PCA". In: *arXiv preprint arXiv:1909.05485* (2019) (cit. on p. 5).

[HW19b] Wen Huang and Ke Wei. "Riemannian Proximal Gradient Methods". In: *arXiv preprint arXiv:1909.06065* (2019) (cit. on p. 1).

[Iva+21] Anastasiya Ivanova, Dmitry Pasechnyuk, Dmitry Grishchenko, Egor Shulgin, Alexander V. Gasnikov, and Vladislav Matyukhin. "Adaptive Catalyst for Smooth Convex Optimization". In: *Optimization and Applications - 12th International Conference, OPTIMA 2021, Petrovac, Montenegro, September 27 - October 1, 2021, Proceedings*. Ed. by Nicholas N. Olenev, Yuri G. Evtushenko, Milojica Jacimovic, Michael Yu. Khachay, and Vlasta Malkova. Vol. 13078. Lecture Notes in Computer Science. Springer, 2021, pp. 20–37 (cit. on p. 2).

[Jou+20] Pooria Joulani, Anant Raj, András György, and Csaba Szepesvári. "A simpler approach to accelerated optimization: iterative averaging meets optimism". In: *Proceedings of the 37th International Conference on Machine Learning, ICML 2020, 13-18 July 2020, Virtual Event*. Vol. 119. Proceedings of Machine Learning Research. PMLR, 2020, pp. 4984–4993 (cit. on p. 2).

[JS21] Jikai Jin and Suvrit Sra. "A Riemannian Accelerated Proximal Extragradient Framework and its Implications". In: *CoRR* abs/2111.02763 (2021) (cit. on p. 5).

[JTU03] Ian T Jolliffe, Nickolay T Trendafilov, and Mudassir Uddin. "A modified principal component technique based on the LASSO". In: *Journal of computational and Graphical Statistics* 12.3 (2003), pp. 531–547 (cit. on p. 1).

[KG20] Dmitry Kamzolov and Alexander Gasnikov. "Near-optimal hyperfast second-order method for convex optimization and its sliding". In: *arXiv preprint arXiv:2002.09050* (2020) (cit. on p. 2).

[Kha+21] Konrawut Khammahawong, Poom Kumam, Parin Chaipunya, and Juan Martínez-Moreno. "Tseng's methods for inclusion problems on Hadamard manifolds". In: *Optimization* (2021), pp. 1–35 (cit. on p. 7).

[KJM19] Hiroyuki Kasai, Pratik Jawanpuria, and Bamdev Mishra. "Riemannian adaptive stochastic gradient algorithms on matrix manifolds". In: *Proceedings of the 36th International Conference on Machine Learning, ICML 2019, 9-15 June 2019, Long Beach, California, USA*. 2019, pp. 3262–3271 (cit. on p. 1).

[KL17] Masoud Badiei Khuzani and Na Li. "Stochastic Primal-Dual Method on Riemannian Manifolds of Bounded Sectional Curvature". In: *16th IEEE International Conference on Machine Learning and Applications, ICMLA 2017, Cancun, Mexico, December 18-21, 2017*. 2017, pp. 133–140 (cit. on p. 2).

[KY22] Jungbin Kim and Insoon Yang. "Nesterov Acceleration for Riemannian Optimization". In: *arXiv preprint arXiv:2202.02036* (2022) (cit. on pp. 1, 2, 5, 6, 8, 22).

[Lev77] Tullio Levi-Civita. *The absolute differential calculus (calculus of tensors)*. Courier Corporation, 1977. ISBN: 978-0-486-31625-3 (cit. on p. 3).

[Lez20] Mario Lezcano-Casado. "Curvature-Dependant Global Convergence Rates for Optimization on Manifolds of Bounded Geometry". In: *arXiv preprint arXiv:2008.02517* (2020) (cit. on p. 4).

[Lin+20] Lizhen Lin, Bayan Saparbayeva, Michael Minyi Zhang, and David B. Dunson. "Accelerated Algorithms for Convex and Non-Convex Optimization on Manifolds". In: *CoRR* abs/2010.08908 (2020) (cit. on p. 5).

[Liu+17] Yuanyuan Liu, Fanhua Shang, James Cheng, Hong Cheng, and Licheng Jiao. "Accelerated First-order Methods for Geodesically Convex Optimization on Riemannian Manifolds". In: *Advances in Neural Information Processing Systems 30: Annual Conference on Neural Information Processing Systems 2017, 4-9 December 2017, Long Beach, CA, USA*. Ed. by Isabelle Guyon, Ulrike von Luxburg, Samy Bengio, Hanna M. Wallach, Rob Fergus, S. V. N. Vishwanathan, and Roman Garnett. 2017, pp. 4868–4877 (cit. on p. 5).

[LM19] Mario Lezcano-Casado and David Martínez-Rubio. "Cheap Orthogonal Constraints in Neural Networks: A Simple Parametrization of the Orthogonal and Unitary Group". In: *Proceedings of the 36th International Conference on Machine Learning, ICML 2019, 9-15 June 2019, Long Beach, California, USA*. 2019, pp. 3794–3803 (cit. on p. 1).

[LMH17] Hongzhou Lin, Julien Mairal, and Zaïd Harchaoui. "Catalyst Acceleration for First-order Convex Optimization: from Theory to Practice". In: *J. Mach. Learn. Res.* 18 (2017), 212:1–212:54 (cit. on p. 2).

[Mar21] David Martínez-Rubio. "Acceleration in first-order optimization methods: promenading beyond convexity or smoothness, and applications". PhD thesis. University of Oxford, 2021 (cit. on p. 2).

[Mar22] David Martínez-Rubio. "Global Riemannian Acceleration in Hyperbolic and Spherical Spaces". In: *International Conference on Algorithmic Learning Theory, 29-1 April 2022, Paris, France*. Ed. by Sanjoy Dasgupta and Nika Haghtalab. Vol. 167. Proceedings of Machine Learning Research. PMLR, 2022, pp. 768–826 (cit. on pp. 4–6, 9, 21).

[MS14] Bamdev Mishra and Rodolphe Sepulchre. "R3MC: A Riemannian three-factor algorithm for low-rank matrix completion". In: *53rd IEEE Conference on Decision and Control, CDC 2014, Los Angeles, CA, USA, December 15-17, 2014*. 2014, pp. 1137–1142 (cit. on p. 1).

[Nes05] Yurii Nesterov. "Smooth minimization of non-smooth functions". In: *Math. Program.* 103.1 (2005), pp. 127–152 (cit. on pp. 4, 6).

[Nes83] Yurii Nesterov. "A method of solving a convex programming problem with convergence rate O(1/k2)". In: *Soviet Mathematics Doklady*. Vol. 27. 1983, pp. 372–376 (cit. on pp. 2, 4).

[Pet06] Peter Petersen. *Riemannian geometry*. Ed. by S Axler and KA Ribet. Vol. 171. Springer, 2006. ISBN: 978-0-387-29403-2 (cit. on p. 2).

[SBC16] Weijie Su, Stephen P. Boyd, and Emmanuel J. Candès. "A Differential Equation for Modeling Nesterov's Accelerated Gradient Method: Theory and Insights". In: *J. Mach. Learn. Res.* 17 (2016), 153:1–153:43 (cit. on p. 2).

| | |
|---|---|
| [SFF19] | Yue Sun, Nicolas Flammarion, and Maryam Fazel. "Escaping from saddle points on Riemannian manifolds". In: *Advances in Neural Information Processing Systems 32: Annual Conference on Neural Information Processing Systems 2019, NeurIPS 2019, 8-14 December 2019, Vancouver, BC, Canada*. 2019, pp. 7274–7284 (cit. on p. 1). |
| [SKM17] | Hiroyuki Sato, Hiroyuki Kasai, and Bamdev Mishra. "Riemannian stochastic variance reduced gradient". In: *CoRR* abs/1702.05594 (2017) (cit. on p. 2). |
| [SKM19] | Hiroyuki Sato, Hiroyuki Kasai, and Bamdev Mishra. "Riemannian Stochastic Variance Reduced Gradient Algorithm with Retraction and Vector Transport". In: *SIAM Journal on Optimization* 29.2 (2019), pp. 1444–1472 (cit. on p. 2). |
| [SQW17] | Ju Sun, Qing Qu, and John Wright. "Complete Dictionary Recovery Over the Sphere II: Recovery by Riemannian Trust-Region Method". In: *IEEE Trans. Inf. Theory* 63.2 (2017), pp. 885–914 (cit. on p. 1). |
| [Tan+14] | Mingkui Tan, Ivor W. Tsang, Li Wang, Bart Vandereycken, and Sinno Jialin Pan. "Riemannian Pursuit for Big Matrix Recovery". In: *Proceedings of the 31th International Conference on Machine Learning, ICML 2014, Beijing, China, 21-26 June 2014*. 2014, pp. 1539–1547 (cit. on p. 1). |
| [TH14] | Guo-ji Tang and Nan-Jing Huang. "Rate of convergence for proximal point algorithms on Hadamard manifolds". In: *Oper. Res. Lett.* 42.6-7 (2014), pp. 383–387 (cit. on p. 7). |
| [Tri+18] | Nilesh Tripuraneni, Nicolas Flammarion, Francis Bach, and Michael I. Jordan. "Averaging Stochastic Gradient Descent on Riemannian Manifolds". In: *CoRR* abs/1802.09128 (2018) (cit. on p. 2). |
| [Tse08] | Paul Tseng. "On accelerated proximal gradient methods for convex-concave optimization". In: *submitted to SIAM Journal on Optimization* 2.3 (2008) (cit. on p. 2). |
| [Van13] | Bart Vandereycken. "Low-Rank Matrix Completion by Riemannian Optimization". In: *SIAM Journal on Optimization* 23.2 (2013), pp. 1214–1236 (cit. on p. 1). |
| [Wal74] | Rolf Walter. "On the metric projection onto convex sets in Riemannian spaces". In: *Archiv der Mathematik* 25.1 (1974), pp. 91–98 (cit. on p. 6). |
| [Wan+15] | Jinhua Wang, Chong Li, Genaro López-Acedo, and Jen-Chih Yao. "Convergence analysis of inexact proximal point algorithms on Hadamard manifolds". In: *J. Glob. Optim.* 61.3 (2015), pp. 553–573 (cit. on p. 7). |
| [Wan+16] | Jinhua Wang, Chong Li, Genaro López-Acedo, and Jen-Chih Yao. "Proximal Point Algorithms on Hadamard Manifolds: Linear Convergence and Finite Termination". In: *SIAM J. Optim.* 26.4 (2016), pp. 2696–2729 (cit. on p. 7). |
| [Wei+16] | Ke Wei, Jian-Feng Cai, Tony F Chan, and Shingyu Leung. "Guarantees of Riemannian optimization for low rank matrix completion". In: *arXiv preprint arXiv:1603.06610* (2016) (cit. on p. 1). |
| [Wie12] | Ami Wiesel. "Geodesic Convexity and Covariance Estimation". In: *IEEE Trans. Signal Process.* 60.12 (2012), pp. 6182–6189 (cit. on p. 1). |
| [WRM16] | Di Wang, Satish Rao, and Michael W. Mahoney. "Unified Acceleration Method for Packing and Covering Problems via Diameter Reduction". In: *43rd International Colloquium on Automata, Languages, and Programming, ICALP 2016, July 11-15, 2016, Rome, Italy*. 2016, 50:1–50:13 (cit. on p. 2). |
| [WS17] | Melanie Weber and Suvrit Sra. "Frank-Wolfe methods for geodesically convex optimization with application to the matrix geometric mean". In: *CoRR* abs/1710.10770 (2017) (cit. on p. 1). |
| [WS19] | Melanie Weber and Suvrit Sra. "Nonconvex stochastic optimization on manifolds via Riemannian Frank-Wolfe methods". In: *CoRR* abs/1910.04194 (2019) (cit. on p. 1). |
| [WWJ16] | Andre Wibisono, Ashia C. Wilson, and Michael I. Jordan. "A Variational Perspective on Accelerated Methods in Optimization". In: *CoRR* abs/1603.04245 (2016) (cit. on p. 2). |
| [ZRS16] | Hongyi Zhang, Sashank J. Reddi, and Suvrit Sra. "Riemannian SVRG: Fast Stochastic Optimization on Riemannian Manifolds". In: *Advances in Neural Information Processing Systems 29: Annual Conference on Neural Information Processing Systems 2016, December 5-10, 2016, Barcelona, Spain*. 2016, pp. 4592–4600 (cit. on p. 2). |
| [ZS16] | Hongyi Zhang and Suvrit Sra. "First-order Methods for Geodesically Convex Optimization". In: *Proceedings of the 29th Conference on Learning Theory, COLT 2016, New York, USA, June 23-26, 2016*. 2016, pp. 1617–1638 (cit. on pp. 1, 2, 6, 9, 21, 24). |

[ZS18]    Hongyi Zhang and Suvrit Sra. "An Estimate Sequence for Geodesically Convex Opti-
          mization". In: *Conference On Learning Theory, COLT 2018, Stockholm, Sweden, 6-9
          July 2018*. Ed. by Sébastien Bubeck, Vianney Perchet, and Philippe Rigollet. Vol. 75.
          Proceedings of Machine Learning Research. PMLR, 2018, pp. 1703–1723 (cit. on pp. 5,
          6, 8).

[ZYF19]   Pan Zhou, Xiao-Tong Yuan, and Jiashi Feng. "Faster First-Order Methods for Stochastic
          Non-Convex Optimization on Riemannian Manifolds". In: *The 22nd International
          Conference on Artificial Intelligence and Statistics, AISTATS 2019, 16-18 April 2019,
          Naha, Okinawa, Japan*. 2019, pp. 138–147 (cit. on p. 1).

[ZZS18]   Jingzhao Zhang, Hongyi Zhang, and Suvrit Sra. "R-SPIDER: A Fast Riemannian
          Stochastic Optimization Algorithm with Curvature Independent Rate". In: *CoRR*
          abs/1811.04194 (2018) (cit. on p. 1).

[ZZS22]   Peiyuan Zhang, Jingzhao Zhang, and Suvrit Sra. "Minimax in Geodesic Metric Spaces:
          Sion's Theorem and Algorithms". In: *CoRR* abs/2202.06950 (2022) (cit. on p. 2).

