# A Optimization lemmas and proofs

We start by noting a property that our parameters satisfy.

**Lemma A.1.** *For the parameter choices of $a_k$ and $A_{k-1}$ in Algorithm 1 we have, for all $k \geq 1$:*

$$\frac{8\lambda}{9}(\xi A_{k-1} + a_k) \geq a_k^2 \geq \frac{3\lambda}{4}(\xi A_{k-1} + \xi a_k).$$

*Proof.* It is a simple computation to check that $a_k$ and $A_{k-1}$ satisfy such inequality. The inequalities are equivalent to the following, which trivially holds:

$$\frac{8}{9}((k^2 - k + 64k\xi - 64\xi + 1000\xi^2) + (2k + 64\xi)) \geq \frac{4}{5}(k^2 + 64k\xi + 1024\xi^2)$$

$$\geq \frac{3}{4}((k^2 - k + 64k\xi - 64\xi + 1000\xi^2) + (2k\xi + 64\xi^2))$$

$\square$

We now prove Proposition 2.1, which will allow us to use $\psi_k$ as a Lyapunov function to show the final convergence rates. The proof will use Lemma A.2 and Lemma A.3, that we state and prove afterwards.

*Proof* (Proposition 2.1). Inequality $(1 - \Delta_k)\psi_k \leq \psi_{k-1}$ is equivalent to

$$(1 - \Delta_k)\left(A_k(f(y_k) - f(x^*)) + \frac{1}{2}\|z_k^{y_k} - x^*\|_{y_k}^2 + \frac{\xi - 1}{2}\|y_k - z_k^{y_k}\|_{y_k}^2\right)$$

$$\leq A_{k-1}(f(y_{k-1}) - f(x^*)) + \left(\frac{1}{2}\|z_{k-1}^{y_{k-1}} - x^*\|_{y_{k-1}}^2 + \frac{\xi - 1}{2}\|y_{k-1} - z_{k-1}^{y_{k-1}}\|_{y_{k-1}}^2\right)$$

which, by bounding $(1 - \Delta_k)(f(y_k) - f(x^*)) \leq f(y_k) - f(x^*)$ and reorganizing, is implied by the following:

$$A_{k-1}(f(y_k) - f(y_{k-1})) + \frac{a_k}{\xi}(f(y_k) - f(x^*)) \leq \frac{1}{2}\|z_{k-1}^{y_{k-1}} - x^*\|_{y_{k-1}}^2 - \frac{1 - \Delta_k}{2}\|z_k^{y_k} - x^*\|_{y_k}^2$$

$$+ \frac{\xi - 1}{2}\left(\|y_{k-1} - z_{k-1}^{y_{k-1}}\|_{y_{k-1}}^2 - (1 - \Delta_k)\|y_k - z_k^{y_k}\|_{y_k}^2\right).$$

We have that due to the projection in Line 12, then $x_k$ is not very far from any $p \in \mathcal{X}$:

$$d(x_k, p) \leq \|x_k - y_{k-1}\|_{y_{k-1}} + d(y_{k-1}, p) \overset{①}{<} \|\bar{z}_{k-1}^{y_{k-1}} - y_{k-1}\|_{y_{k-1}} + D \overset{②}{\leq} 2D, \qquad (1)$$

where ① holds by the definition of $x_k$ and the fact $y_{k-1}, p \in \mathcal{X}$, and ② is due to the projection defining $\bar{z}_{k-1}^{y_{k-1}}$. Now we use the first part of Lemma A.2 with both $x \leftarrow y_{k-1}$ and $x \leftarrow x^*$ and we bound the resulting errors $\varepsilon_k(\cdot)$ by using the second part of Lemma A.2. We also use Lemma A.3, so it is enough to prove the following:

$$A_{k-1}\langle v_k^x, x_k - y_{k-1}\rangle + (a_k/\xi)\langle v_k^x, x_k - z_{k-1}^{x_k} + z_{k-1}^{x_k} - x^*\rangle - \frac{4\lambda}{9}(A_{k-1} + a_k/\xi)\|v_k^x\|^2$$

$$\leq \frac{1}{2}\|z_{k-1}^{x_k} - x^*\|_{x_k}^2 - \frac{1}{2}\|z_k^{x_k} - x^*\|_{x_k}^2 + \frac{\xi - 1}{2}\left(\|x_k - z_{k-1}^{x_k}\|_{x_k}^2 - \|x_k - z_k^{x_k}\|_{x_k}^2\right),$$

Note that thanks to Lemma A.3 now we have the potentials on the right hand side as expressions in the tangent space of $x_k$. Also, note that we have canceled some potentials proportional to $\Delta_k$ coming from the bound on the error $\varepsilon_k(\cdot)$. Now we use that by definition of $x_k$ we have, for all $v \in T_{x_k}\mathcal{M}$, $A_{k-1}\langle v, x_k - y_{k-1}\rangle = -a_k\langle v, x_k - z_{k-1}^{x_k}\rangle$, so we use this fact for $v = v_k^x$ and use the following identity, that holds by the definion of $z_k^{x_k} \overset{\text{def}}{=} z_{k-1}^{x_k} - \eta_k v_k^x$:

$$\frac{a_k/\xi}{\eta_k}\langle \eta_k v_k^x, z_{k-1}^{x_k} - x^*\rangle = \frac{a_k/\xi}{2\eta_k}\left(\eta_k^2\|v_k^x\|_{x_k}^2 + \|z_{k-1}^{x_k} - x^*\|_{x_k}^2 - \|z_k^{x_k} - x^*\|_{x_k}^2\right).$$

so that, after canceling terms, it is enough to prove:

$$
a_k(1 - 1/\xi)\langle -v_k^x, x_k - z_{k-1}^{x_k}\rangle - \frac{a_k(1 - 1/\xi)}{2\eta_k}\eta_k^2\|v_k^x\|^2
$$
$$
+ \|v_k^x\|^2(-\frac{4}{9}(A_{k-1}\lambda + a_k\lambda/\xi) + \frac{a_k\eta_k}{2}) \tag{2}
$$
$$
\leq \frac{\xi - 1}{2}\left(\|x_k - z_{k-1}^{x_k}\|_{x_k}^2 - \|x_k - z_k^{x_k}\|_{x_k}^2\right),
$$

Now we show that in the previous inequality (2), the first line cancels with the last line. Note that $(a_k(1 - 1/\xi))/\eta_k = (1 - 1/\xi)/(1/\xi) = \xi - 1$. Thus, by using again the definition of $z_k^{x_k}$, we have:

$$
\frac{a_k(1 - 1/\xi)}{\eta_k}\langle -\eta_k v_k^x, x_k - z_{k-1}^{x_k}\rangle = \frac{a_k(1 - 1/\xi)}{2\eta_k}\left(\eta_k^2\|v_k^x\|_{x_k}^2 + \|x_k - z_{k-1}^{x_k}\|_{x_k}^2 - \|x_k - z_k^{x_k}\|_{x_k}^2\right).
$$

Finally, it only remains to prove:

$$
\frac{\|v_k^x\|^2}{2\xi} \cdot \left(-\frac{8}{9}(\xi A_{k-1}\lambda + a_k\lambda) + a_k^2\right) \leq 0, \tag{3}
$$

which holds by Lemma A.1. $\qquad\square$

We now show the two auxiliary lemmas that we used in the previous proof.

**Lemma A.2.** *Let $h_k(x) \overset{\text{def}}{=} f(x) + \frac{1}{2\lambda}d(x_k, x)^2$ be the strongly g-convex function used at step $k$, and let $y_k^* = \arg\min_{y\in\mathcal{X}} h_k(y)$. Then, for $y_k \in \mathcal{X}$, if we let $v_k^x \overset{\text{def}}{=} -\text{Log}_{x_k}(y_k)/\lambda$, then the following holds, for all $x \in \mathcal{X}$:*

$$
f(x) \geq f(y_k) + \langle v_k^x, x - x_k\rangle_{x_k} + \frac{\lambda}{2}\|v_k^x\|^2 - \varepsilon_k(x)
$$

*where $\varepsilon_k(x) \overset{\text{def}}{=} -\frac{1}{\lambda}\langle y_k - y_k^*, x - x_k\rangle_{x_k} + (h_k(y_k) - h_k(y_k^*))$. Moreover, if $y_k$ satisfies*

$$
h_k(y_k) - h_k(y_k^*) \leq \frac{\Delta_k d(x_k, y_k^*)^2}{78\lambda},
$$

*then we have*

$$
-\frac{\lambda}{2}\|v_k^x\|^2(A_{k-1} + a_k/\xi) + a_k\varepsilon_k(x^*)/\xi + A_{k-1}\varepsilon_k(y_{k-1})
$$
$$
\leq -\frac{4\lambda\|v_k^x\|^2}{9}(A_{k-1} + a_k/\xi) + \frac{\Delta_k}{2}\left(\|x^* - z_{k-1}^{x_k}\|_{x_k}^2 + (\xi - 1)\|x_k - z_{k-1}^{x_k}\|_{x_k}^2\right).
$$

*Proof.* The function $h_k$ is $\frac{1}{\lambda}$-strongly g-convex because by Fact 1.3 the function $\frac{1}{2}d(x_k, x)^2$ is 1-strongly g-convex in a Hadamard manifold. By the first-order optimality condition of $h_k$ at $y_k^*$ we have that $\tilde{v}_k^y \overset{\text{def}}{=} \lambda^{-1}\text{Log}_{y_k^*}(x_k) \in \partial(f + I_{\mathcal{X}})(y_k^*)$ is a subgradient of $f + I_{\mathcal{X}}$ at $y_k^*$. Thus, we have, for all $x \in \mathcal{X}$ and for $\tilde{v}_k^x \overset{\text{def}}{=} \Gamma_{y_k^*}^{x_k}(\tilde{v}_k^y)$:

$$
f(x) \overset{①}{\geq} f(y_k^*) + \langle \tilde{v}_k^y, x - y_k^*\rangle_{y_k^*}
$$
$$
\overset{②}{\geq} f(y_k^*) + \langle \tilde{v}_k^x, x - x_k\rangle_{x_k} + \lambda\|\tilde{v}_k^x\|^2
$$
$$
\overset{③}{=} f(y_k) + \langle v_k^x, x - x_k\rangle_{x_k} + \frac{\lambda}{2}\|v_k^x\|^2 + \frac{\lambda}{2}\|\tilde{v}_k^x\|^2
$$
$$
+ \langle \tilde{v}_k^x - v_k^x, x - x_k\rangle_{x_k} + \left((f(y_k^*) + \frac{\lambda}{2}\|\tilde{v}_k^x\|^2) - (f(y_k) + \frac{\lambda}{2}\|v_k^x\|^2)\right)
$$
$$
\overset{④}{\geq} f(y_k) + \langle v_k^x, x - x_k\rangle_{x_k} + \frac{\lambda}{2}\|v_k^x\|^2 + \frac{1}{\lambda}\langle y_k - y_k^*, x - x_k\rangle_{x_k} - (h_k(y_k) - h_k(y_k^*))
$$

where ① holds because $\tilde{v}_k^y \in \partial(f + I_{\mathcal{X}})(y_k^*)$ and $x, y_k^* \in \mathcal{X}$. In ②, we used the first part of Lemma B.5 along with $\delta = 1$. We just added and subtracted some terms in ③, and in ④, we dropped $\frac{\lambda}{2}\|\tilde{v}_k^x\|^2$, and we used the definitions of $h_k$, $\tilde{v}_k^x$, and $v_k^x = -\mathrm{Log}_{x_k}(y_k)/\lambda$.

Now we proceed to prove the second part. The following holds:

$$
-\frac{a_k}{\lambda\xi}\langle y_k - y_k^*, x^* - x_k\rangle_{x_k} - A_{k-1}\frac{1}{\lambda}\langle y_k - y_k^*, y_{k-1} - x_k\rangle_{x_k}
$$
$$
\overset{①}{\le} \frac{1}{\lambda}\|y_k - y_k^*\|_{x_k} \cdot \|\frac{a_k}{\xi}x^* + A_{k-1}y_{k-1} - (\frac{a_k}{\xi} + A_{k-1})x_k\|_{x_k}
$$
$$
\overset{②}{\le} \frac{1}{\lambda}d(y_k, y_k^*) \cdot \frac{a_k}{\xi}\|x^* - z_{k-1}^{x_k} + (\xi-1)(x_k - z_{k-1}^{x_k})\|_{x_k}
$$
$$
\overset{③}{\le} \frac{1}{\lambda}\sqrt{2\lambda(h_k(y_k) - h_k(y_k^*))} \cdot \frac{a_k}{\xi}\sqrt{\xi}\sqrt{\|x^* - z_{k-1}^{x_k}\|_{x_k}^2 + (\xi-1)\|(x_k - z_{k-1}^{x_k})\|_{x_k}^2} \tag{4}
$$
$$
= \sqrt{\frac{2a_k^2(h_k(y_k) - h_k(y_k^*))}{\Delta_k\lambda\xi}} \cdot \sqrt{\Delta_k}\sqrt{\|x^* - z_{k-1}^{x_k}\|_{x_k}^2 + (\xi-1)\|(x_k - z_{k-1}^{x_k})\|_{x_k}^2}
$$
$$
\overset{④}{\le} \frac{a_k^2(h_k(y_k) - h_k(y_k^*))}{\Delta_k\lambda\xi} + \frac{\Delta_k}{2}(\|x^* - z_{k-1}^{x_k}\|_{x_k}^2 + (\xi-1)\|(x_k - z_{k-1}^{x_k})\|_{x_k}^2),
$$

where ① groups some terms and uses Cauchy-Schwartz. In inequality ②, for the first term we bounded the distance between $y_k^*$ and $y_k$ estimated from $T_{x_k}\mathcal{M}$ by the actual distance, which is a property that holds in Hadamard manifolds and it holds by the first part of Corollary B.2 with $\delta = 1$, $p \leftarrow y_k^*$, $y \leftarrow y_k$, $x \leftarrow x_k$, $z^y \leftarrow 0$. The second term is substituted by a term of equal value by using Euclidean trigonometry in $T_{x_k}\mathcal{M}$, as in the following. Let $w \overset{\text{def}}{=} \frac{1}{a_k/\xi + A_{k-1}}(\frac{a_k}{\xi}\mathrm{Log}_{x_k}(x^*) + A_{k-1}\mathrm{Log}_{x_k}(y_{k-1}))$ and let $u \in T_{x_k}$ be the point in the line containing $\mathrm{Log}_{x_k}(y_{k-1})$ and $0 = \mathrm{Log}_{x_k}(x_k) \in T_{x_k}$ such that the triangle with vertices $0$, $\mathrm{Log}_{x_k}(y_{k-1})$ and $w$ and the triangle with vertices $u$, $\mathrm{Log}_{x_k}(y_{k-1})$ and $\mathrm{Log}_{x_k}(x^*)$ are similar triangles, and so

$$
\frac{\|\mathrm{Log}_{x_k}(x^*) - u\|}{\|w - \mathrm{Log}_{x_k}(x_k)\|} \overset{⑤}{=} \frac{\|\mathrm{Log}_{x_k}(x^*) - \mathrm{Log}_{x_k}(y_{k-1})\|}{\|w - \mathrm{Log}_{x_k}(y_{k-1})\|} \overset{⑥}{=} \frac{A_{k-1} + a_k/\xi}{a_k/\xi}. \tag{5}
$$

We used the triangle similarity in ⑤ and in ⑥ we used the definition of $w$ as a convex combination of $\mathrm{Log}_{x_k}(x^*)$ and $\mathrm{Log}_{x_k}(y_{k-1})$. It is enough to show $u = \xi z_{k-1}^{x_k}$ as in such a case what we applied in ② is equivalent to the equality (5) above. By the definition of $x_k$, we have ⑦ below and by triangle similarity we have ⑧ below:

$$
z_{k-1}^{x_k} \overset{⑦}{=} -\frac{A_{k-1}}{a_k}\mathrm{Log}_{x_k}(y_{k-1}) \overset{⑧}{=} \frac{A_{k-1}}{a_k} \cdot \frac{a_k/\xi}{A_{k-1}}u = \frac{1}{\xi}u,
$$

as desired. In the next inequality ③, we used that by $(1/\lambda)$-strong g-convexity of $h_k$ and by optimality of $y_k^*$, we have $\frac{1}{2\lambda}d(\cdot, y_k^*)^2 \le h_k(\cdot) - h_k(y_k^*)$. For the second term, we used that for vectors $b, c \in \mathbb{R}^n$ and $\omega \in \mathbb{R}_{\geq 0}$, we have, by Young's inequality, $\|b + wc\| = \sqrt{\|b\|^2 + \omega^2\|c\|^2 + 2\langle\sqrt{\omega}b, \sqrt{\omega}c\rangle} \le \sqrt{(1+\omega)(\|b\|^2 + \omega\|c\|^2)}$. In ④ we used Young's inequality.

Before we conclude, we note that

$$
d(x_k, y_k^*) \le \sqrt{2}d(x_k, y_k), \tag{6}
$$

which is implied by the following, where we use the same as in ③ above, the assumption on $y_k$ and $\Delta_k \le 1$:

$$
d(x_k, y_k^*) \le d(x_k, y_k) + d(y_k, y_k^*) \le d(x_k, y_k) + \sqrt{2\lambda(h_k(y_k) - h_k(y_k^*))}
$$
$$
\le d(x_k, y_k) + \sqrt{\Delta_k/34} \cdot d(x_k, y_k^*) \le d(x_k, y_k) + d(x_k, y_k^*)/4.
$$

Finally, we can make use of (4) and (6) to obtain the claim in the second part of the lemma:

$$
-\frac{\lambda}{2}\|v_k^x\|^2(A_{k-1}+a_k/\xi)+a_k\varepsilon_k(x^*)/\xi+A_{k-1}\varepsilon_k(y_{k-1})-\frac{\Delta_k}{2}\|x^*-z_{k-1}^{x_k}\|_{x_k}^2
$$
$$
-\Delta_k\frac{\xi-1}{2}\|(x_k-z_{k-1}^{x_k})\|_{x_k}^2
$$
$$
\leq -\frac{\lambda}{2}\|v_k^x\|^2(A_{k-1}+a_k/\xi)+\left(A_{k-1}+a_k/\xi+\frac{a_k^2}{\Delta_k\lambda\xi}\right)(h_k(y_k)-h_k(y_k^*))
$$
$$
\overset{①}{\leq} -\frac{\lambda}{2}\|v_k^x\|^2(A_{k-1}+a_k/\xi)+(A_{k-1}+a_k/\xi)\left(1+\frac{a_k^2}{(\xi A_{k-1}+a_k)\lambda}\right)\frac{d(x_k,y_k)^2}{34\lambda}
$$
$$
\overset{②}{\leq} -\frac{\lambda}{2}\|v_k^x\|^2(A_{k-1}+a_k/\xi)+\frac{d(x_k,y_k)^2}{18\lambda}(A_{k-1}+a_k/\xi)
$$
$$
\overset{③}{=} -\frac{4\lambda\|v_k^x\|^2}{9}(A_{k-1}+a_k/\xi),
$$

where ① holds by the assumption on $y_k$, $\Delta_k\leq 1$, and (6). Inequality ② uses the upper bound on $a_k^2$ in Lemma A.1, and ③ uses the definition $v_k^x\overset{\text{def}}{=}-\text{Log}_{x_k}(y_k)/\lambda$.

□

The following lemma allows to *move* the regularized lower bounds on the objective without incurring extra geometric penalties.

**Lemma A.3 (Translating Potentials with no Geometric Penalty).** *Using the variables in Algorithm 1, for any $\Delta_k\in[0,1)$, we have*

$$
\|z_{k-1}^{x_k}-x^*\|_{x_k}^2-(1-\Delta_k)\|z_k^{x_k}-x^*\|_{x_k}^2+(\xi-1)\left(\|x_k-z_{k-1}^{x_k}\|_{x_k}^2-(1-\Delta_k)\|x_k-z_k^{x_k}\|_{x_k}^2\right)
$$
$$
\leq \|z_{k-1}^{y_{k-1}}-x^*\|_{y_{k-1}}^2-(1-\Delta_k)\|z_k^{y_k}-x^*\|_{y_k}^2
$$
$$
+(\xi-1)\left(\|y_{k-1}-z_{k-1}^{y_{k-1}}\|_{y_{k-1}}^2-(1-\Delta_k)\|y_k-z_k^{y_k}\|_{y_k}^2\right).
$$

*Proof.* Firstly, by the projection step in Line 12, we have

$$
\|z_{k-1}^{y_{k-1}}-x^*\|_{y_k}^2\geq\|\bar{z}_{k-1}^{y_{k-1}}-x^*\|_{y_k}^2\qquad\text{and}\qquad(\xi-1)\|z_{k-1}^{y_{k-1}}\|_{y_k}^2\geq(\xi-1)\|\bar{z}_{k-1}^{y_{k-1}}\|_{y_k}^2\qquad(7)
$$

since the operation is a simple Euclidean projection onto the closed ball $\bar{B}(0,D)$ in $T_{y_k}\mathcal{M}$. By the second part of Corollary B.2, $y=x_k$ and $x=y_{k-1}$ and by (1), we have ① below

$$
\|\bar{z}_{k-1}^{y_{k-1}}-x^*\|_{y_{k-1}}^2+(\xi-1)\|\bar{z}_{k-1}^{y_{k-1}}\|_{y_{k-1}}^2\overset{①}{\geq}\|z_{k-1}^{x_k}-x^*\|_{x_k}^2+(\zeta_{2D}-1)\|z_{k-1}^{x_k}\|_{x_k}^2+(\xi-\zeta_{2D})\|\bar{z}_{k-1}^{y_{k-1}}\|_{y_{k-1}}^2
$$
$$
\overset{②}{\geq}\|z_{k-1}^{x_k}-x^*\|_{x_k}^2+(\xi-1)\|z_{k-1}^{x_k}\|_{x_k}^2+(\xi-\zeta_{2D})\left(\left(\frac{A_{k-1}+a_k}{A_{k-1}}\right)^2-1\right)\|z_{k-1}^{x_k}\|_{x_k}^2
$$
$$
\overset{③}{\geq}\|z_{k-1}^{x_k}-x^*\|_{x_k}^2+(\xi-1)\|z_{k-1}^{x_k}\|_{x_k}^2+\frac{3(\xi-1)}{2}\left(\frac{1}{1-\tau_k}-1\right)\|z_{k-1}^{x_k}\|_{x_k}^2,
$$
(8)

and ② uses the definition of $x_k$. In ③, we used the definition of $\xi=4\zeta_{2D}-3$ that implies $\xi-\zeta_{2D}\geq\frac{3}{4}(\xi-1)$ and for $\tau_k\overset{\text{def}}{=}a_k/(a_k+A_{k-1})$ we have that $(1+\frac{a_k}{A_{k-1}})^2-1\geq\frac{2a_k}{A_{k-1}}=2(\frac{1}{1-\tau_k}-1)$. Now, using the second part of Lemma B.1 with $y=y_k$, $x=x_k$ $z^x=-\eta_k v_k^x$, $a^x=z_{k-1}^{x_k}$, so that $z^x+a^x=z_k^{x_k}$ and $z^y+a^y=z_k^{y_k}$ and

$$
r=\frac{\|\text{Log}_{x_k}(y_k)\|}{\|z^x\|}=\frac{\lambda\|v_k^x\|}{\eta_k\|v_k^x\|}=\frac{\xi\lambda}{a_k}=\frac{5\xi}{2k+64\xi}<5/6<1.\qquad(9)
$$

Note that by the choice of parameters and the fact that $r < 1$, the assumptions in Lemma B.1 are satisfied. Thus, the following holds

$$\|z_k^{x_k} - x^*\|_{x_k}^2 + (\xi - 1)\|z_k^{x_k}\|_{x_k}^2 + \frac{\xi - 1}{2}\left(\frac{r}{1-r}\right)\|z_{k-1}^{x_k}\|^2 \geq \|z_k^{y_k} - x^*\|_{y_k}^2 + (\xi - 1)\|z_k^{y_k}\|_{y_k}^2. \tag{10}$$

Hence, combining (7), (8) and (10) we obtain that it is enough to prove

$$-(1 - \Delta_k)\left(\frac{r}{1-r}\right) + 3\left(\frac{1}{1-\tau_k} - 1\right) \geq 0,$$

The proof will be finished if we prove the result for $\Delta_k = 0$. If we use this last inequality, and the fact that for $r \leq 5/6$, we have $\frac{r}{1-r} \leq 3\left(\frac{1}{1-3r/4} - 1\right)$, we deduce that it suffices to show $\tau_k \geq \frac{3}{4}r$ to conclude

$$\frac{r}{1-r} \leq 3\left(\frac{1}{1-3r/4} - 1\right) \leq 3\left(\frac{1}{1-\tau_k} - 1\right).$$

Such inequality, namely $\tau_k \geq \frac{3}{4}r$, is equivalent to $\frac{a_k^2}{\lambda} \geq \frac{3\xi}{4}(a_k + A_{k-1})$ and it holds by Lemma A.1. $\square$

Algorithm 1 employs a linearly convergent RGD as a subroutine in order to compute Line 8. Below, we show how this is done and we note that any other linearly convergent algorithm can be used to solve this step. We first describe a warm start that we will use for RGD. The warm start allows to know when to stop the subroutine at the same time that it will guarantee fast convergence. One should think about this lemma as being applied to $h_k(\cdot) \stackrel{\text{def}}{=} f(\cdot) + \frac{1}{2\lambda}d(\cdot, x_k)^2$. Also, note that in that case we can compute the gradient of $h$ at any point $y \in \mathcal{X}$ as $\nabla h(y) = \nabla f(y) + \frac{1}{\lambda}\text{Log}_y(x_k)$.

**Lemma A.4 (Warm start).** *Let $\mathcal{M}$ be a Hadamard manifold, let $x \in \mathcal{M}$, $\mathcal{X} \subset \mathcal{M}$ be a uniquely geodesic convex set of diameter $D$ and $h : \mathcal{M} \to \mathbb{R}$ a geodesically convex and $L'$-smooth function. Assume access to a projection operator $\mathcal{P}_{\mathcal{X}}$ on $\mathcal{X}$. Let $x' = \mathcal{P}_{\mathcal{X}}(x)$ and $x^+ \stackrel{\text{def}}{=} \text{Exp}_{x'}(-\frac{1}{L'}\nabla h(x'))$ and $p_0 \stackrel{\text{def}}{=} \mathcal{P}_{\mathcal{X}}(x^+)$ and $D' \stackrel{\text{def}}{=} d(x^+, x') = \|\nabla h(x')\|/L'$. We have that, for all $p \in \mathcal{X}$:*

$$h(p_0) - h(p) \leq \frac{\zeta_{D'}L'}{2}d(x', p)^2 \leq \frac{\zeta_{D'}L'}{2}d(x, p)^2.$$

*Proof.* With the notation of the lemma, we have, by smoothness of $h$, that the following quadratic $Q : T_{x'}\mathcal{M} \to \mathbb{R}$, $v \mapsto h(x') + \frac{L'}{2}\|x^+ - v\|_{x'}^2 - \frac{L'}{2}\|x^+ - x'\|_{x'}^2$ induces an upper bound on $h$ in $\mathcal{X}$, via $\text{Exp}_{x'}(\cdot)$. Thus, we have

$$-\frac{\zeta_{D'}L'}{2}d(x, p)^2 + h(p_0) \overset{①}{\leq} -\frac{\zeta_{D'}L'}{2}d(x', p)^2 + h(p_0)$$

$$\overset{②}{\leq} -\frac{\zeta_{D'}L'}{2}d(x', p)^2 + Q(\text{Log}_{x'}(p_0))$$

$$\overset{③}{\leq} -\frac{\zeta_{D'}L'}{2}d(x', p)^2 + \left(h(x') + \frac{L'}{2}d(x^+, p_0)^2 - \frac{L'}{2}d(x^+, x')^2\right)$$

$$\overset{④}{\leq} -\frac{\zeta_{D'}L'}{2}d(x', p)^2 + \left(h(x') + \frac{L'}{2}d(x^+, p)^2 - \frac{L'}{2}d(x^+, x')^2\right)$$

$$\overset{⑤}{\leq} -L'\langle\text{Log}_{x'}(p), \text{Log}_{x'}(x^+)\rangle + h(x')$$

$$\overset{⑥}{=} -L'\langle\text{Log}_{x'}(p), -\frac{1}{L'}\nabla h(x')\rangle + h(x')$$

$$\overset{⑦}{\leq} h(p).$$

We used the projection property of $x' = \mathcal{P}_{\mathcal{X}}(x)$ in ①. We used smoothness in ②. In ③, we used the first part of Corollary B.2 with $\delta_{D'} = 1$, $r = 1$, $x \leftarrow x'$, $y \leftarrow x^+$, $p \leftarrow p_0$ to bound the

814 estimated distance $\|x^+ - p_0\|_{x'}$ by the actual distance $d(x^+, p_0)$. We used the projection property
815 of $p_0 = \mathcal{P}_{\mathcal{X}}(x^+)$ in ④. In ⑤, we used the version of Corollary B.3 in Remark B.4. We used the
816 definition of $x^+$ in ⑥, and we conclude in ⑦ by using g-convexity of $h$. $\qquad\square$

817 Here we finish the computations of the reasoning in Remark 2.3.

818 **Remark A.5.** *Let* $D'' \overset{\text{def}}{=} (L_{f,\mathcal{X}} + 2LD/\zeta_{2D})/L'$, *where* $L_{f,\mathcal{X}}$ *is the Lipschitz constant of*
819 *$f$ in $\mathcal{X}$. If we initialize the projected RGD method in [ZS16, Theorem 15] with* $p_0 \overset{\text{def}}{=}$
820 *$\mathcal{P}_{\mathcal{X}}(\mathrm{Exp}_{x'_k}(-\frac{1}{L'}\nabla h_k(x'_k)))$, where $x'_k \overset{\text{def}}{=} \mathcal{P}_{\mathcal{X}}(x_k)$, then using $(L/\zeta_{2D})$-strong g-convexity of $h_k$ to*
821 *bound* $L'd(p_0, y_k^*)^2 \leq 4\zeta_{2D}(h_k(p_0) - h(y_k^*))$, *using Lemma A.4 with* $x \leftarrow x_k$, $p \leftarrow y_k^*$,

$$D' \leftarrow \|\nabla h_k(x')\|/L' \leq (\|\nabla f(x')\| + L\|\mathrm{Log}_{x_k}(x')\|/\zeta_{2D})/L' \leq (L_{f,\mathcal{X}} + 2LD/\zeta_{2D})/L' = D'',$$

822 *and using the guarantees on $\mathcal{A}$, we have that we find a point $y_k$ satisfying $h_k(y_k) - h_k(y_k^*) \leq$*
823 *$\frac{\Delta_k d(x_k, y_k^*)^2}{78\lambda}$ in $\widetilde{O}(\zeta)$ queries to the gradient and projection oracles. Indeed, the number of queries is*
824 *given by*

$$O\left(\zeta_{2D} \log \frac{(h_k(p_0) - h_k(y_k^*)) + L'd(p_0, y_k^*)^2}{\Delta_k d(x_k, y_k^*)^2/(78\zeta_{2D}/L)}\right) = O\left(\zeta \log \frac{78\zeta \cdot (1 + 4\zeta_{2D})(\zeta_{D'}L'/2)d(x_k, y_k^*)^2}{L\Delta_k d(x_k, y_k^*)^2}\right)$$
$$= O\left(\zeta \log \left(\frac{\zeta \cdot \zeta_{D'}}{\Delta_k}\right)\right) = O\left(\zeta \log \left(\frac{\zeta \cdot \zeta_{D''}}{\Delta_k}\right)\right).$$

825 *Note we know that on the one hand we can stop the algorithm after $O(\zeta \log(\frac{\zeta \cdot \zeta_{D'}}{\Delta_k}))$ iterations which*
826 *is a value we can compute, including constants, since we can compute $D'$. On the other hand the*
827 *worst-case complexity can be expressed as $O(\zeta \log(\frac{\zeta \cdot \zeta_{D''}}{\Delta_k}))$ but we do not need to have access to*
828 *$L_{f,\mathcal{X}}/L'$. Note that if there is a point $x^* \in \mathcal{X}$ such that $\nabla f(x^*) = 0$, then we have by smoothness*
829 *that $L_{f,\mathcal{X}} = O(LD)$ and therefore $D'' = O(D)$.*

830 Finally, we use Proposition 2.1 and Remark 2.3 to show the final convergence rates for g-convex
831 functions.

832 *Proof* (Theorem 2.2). Given the inequality $(1 - \Delta_k)\psi_k \leq \psi_{k-1}$, proven in Proposition 2.1, we
833 can use $\psi_k$ as a Lyapunov function in order to prove convergence rates of Algorithm 1. It follows
834 straightforwardly by definition of $\psi_k$, in the following way

$$f(y_k) - f(x^*) \leq \frac{\psi_k}{A_k} \leq \prod_{i=1}^{k}(1 - \Delta_i)^{-1}\frac{\psi_0}{A_k} \overset{①}{\leq} \frac{2\psi_0}{A_k} \overset{②}{\leq} 2LR_0^2\left(\frac{A_0}{A_k} + \frac{1}{4LA_k}\right)$$
$$= O\left(LR_0^2\left(\frac{\lambda\xi}{\lambda\left(\frac{k^2+\xi k}{\xi} + \xi\right)} + \frac{1}{\lambda L\left(\frac{k^2+\xi k}{\xi} + \xi\right)}\right)\right)$$
$$= O\left(LR_0^2\left(\frac{\xi^2}{k^2 + \xi k + \xi^2}\right)\right) \overset{③}{=} O\left(\frac{LR_0^2}{k^2} \cdot \zeta^2\right).$$

835 In ①, we used $\prod_{i=1}^{k}(1 - \Delta_k) = \prod_{i=1}^{k}\frac{i(i+2)}{(i+1)^2} = \frac{k+2}{2(k+1)} \geq \frac{1}{2}$. We used smoothness in ②. Note
836 $\frac{\xi-1}{2}\|y_0 - z_0^{y_0}\|_{y_0} = 0$ and $\|z_0^{y_0} - x^*\|_{y_0}^2 = R_0^2$. In ③, we used $\xi = O(\zeta)$ and we dropped some terms
837 in the denominator. Secondly, since the computation of the approximate proximal operator takes
838 $\widetilde{O}(\zeta)$ queries to the gradient and projection oracle, cf. Remark 2.3, and $\Delta_k^{-1} \leq \Delta_T^{-1} = (T + 1)^2$,
839 then the total number of queries made to these oracles to obtain an $\varepsilon$-minimizer is bounded by
840 $\widetilde{O}\left(\zeta^2\sqrt{\frac{LR_0^2}{\varepsilon}}\right).$ $\qquad\square$

841 We present now the proof that yields an accelerated algorithm for strongly g-convex and smooth
842 functions.

843 *Proof* (Theorem 2.4). The statement of the reduction in [Mar22, Theorem 7] assumes a function
844 $f : \mathcal{M} \to \mathbb{R}$ to be optimized has a global minimizer in an unconstrained problem, but the same proof
845 of this theorem works if we have a $\mu$-strongly g-convex and $L$-smooth function $f$ defined over an open

set containing a closed geodesically convex set $\mathcal{X}$ and a minimizer $x^*$ of this function restricted to $\mathcal{X}$. The reduction provides an algorithm for optimizing $f$ by using $O(\mathrm{Time}_{\mathrm{ns}}(L, \mu, R) \log(\mu R^2/\varepsilon))$ queries to the oracle, where $\mathrm{Time}_{\mathrm{ns}}(L, \mu, R)$ is the number of times the oracle is queried by the non-strongly g-convex algorithm if the initial distance is upper bounded by $R$ and if we require accuracy $\mu R^2/4$. In our case, it is $\mathrm{Time}_{\mathrm{ns}}(L, \mu, R) = O(\zeta^2 \log(\zeta^2 \sqrt{L/\mu})\sqrt{\frac{L}{\mu}}) = O^*(\zeta^2 \sqrt{\frac{L}{\mu}})$, so the result follows. We note that the reverse reduction yields extra geometric penalties but this one does not. $\qquad\square$

## B  Geometric lemmas

In this section, we state and prove Lemma B.5, which is used in the proof of Theorem 2.2 to show that the lower bound given by $f(y_k^*) + \langle \tilde{v}_k^y, x - y_k^* \rangle$ that is affine if pulled-back to $T_{y_k^*}$ can be bounded by another function, that is affine if pulled back to $x_k$. We also include and prove, with some generalizations, some known Riemannian inequalities that are used in Riemannian optimization methods and that we also use. The second part of the following lemma appeared in [KY22]. Similarly with the second part of the corollary that follows.

In this section, unless otherwise specified, $\mathcal{M}$ is an $n$-dimensional Riemannian manifold of bounded sectional curvature.

**Lemma B.1.** *Let $x, y, p \in \mathcal{M}$ be the vertices of a uniquely geodesic triangle $\mathcal{T}$ of diameter $D$, and let $z^x \in T_x\mathcal{M}$, $z^y \stackrel{\mathrm{def}}{=} \Gamma_x^y(z^x) + \mathrm{Log}_y(x)$, such that $y = \mathrm{Exp}_x(rz^x)$ for some $r \in [0, 1)$. If we take vectors $a^y \in T_y\mathcal{M}$, $a^x \stackrel{\mathrm{def}}{=} \Gamma_y^x(a^y) \in T_x\mathcal{M}$, then we have the following, for all $\xi \geq \zeta_D$:*

$$\|z^y + a^y - \mathrm{Log}_y(p)\|_y^2 + (\delta_D - 1)\|z^y + a^y\|_y^2$$
$$\geq \|z^x + a^x - \mathrm{Log}_x(p)\|_x^2 + (\delta_D - 1)\|z^x + a^x\|_x^2 - \frac{\xi - \delta_D}{2}\left(\frac{r}{1-r}\right)\|a^x\|_x^2,$$

*and*

$$\|z^y + a^y - \mathrm{Log}_y(p)\|_y^2 + (\xi - 1)\|z^y + a^y\|_y^2$$
$$\leq \|z^x + a^x - \mathrm{Log}_x(p)\|_x^2 + (\xi - 1)\|z^x + a^x\|_x^2 + \frac{\xi - \delta_D}{2}\left(\frac{r}{1-r}\right)\|a^x\|_x^2.$$

*Proof.* Let $\gamma$ be the unique geodesic in $\mathcal{T}$ such that $\gamma(0) = x$ and $\gamma(r) = y$. We have $\gamma'(0) = z^x$. Along $\gamma$, we define the vector field $V(t) = \Gamma_0^t(\gamma)(z^x - t\gamma'(0))$. Then, it is $V'(t) = -\gamma'(t)$, and $\|V(t)\| = \|a + (1-t)z^x\|$. We will make use of the potential $w : [0, r] \to \mathbb{R}$ defined as $w(t) = \|\mathrm{Log}_{\gamma(t)}(x) - V(t)\|^2$. We can compute

$$\begin{aligned}
\frac{d}{dt}w(t) &= 2\langle D_t(\mathrm{Log}_{\gamma(t)}(x) - V(t)), \mathrm{Log}_{\gamma(t)}(x) - V(t)\rangle \\
&= 2\langle D_t \mathrm{Log}_{\gamma(t)}(x), \mathrm{Log}_{\gamma(t)}(x)\rangle - 2\langle D_t \mathrm{Log}_{\gamma(t)}(x), V(t)\rangle \\
&\quad - 2\langle D_t V(t), \mathrm{Log}_{\gamma(t)}(x)\rangle + 2\langle D_t V(t), V(t)\rangle \\
&= -2\langle D_t(\mathrm{Log}_{\gamma(t)}(x), V(t)\rangle + 2\langle D_t V(t), V(t)\rangle.
\end{aligned} \tag{11}$$

Now, we bound the first summand. We use that for the function $\Phi_p(x) = \frac{1}{2}d(x, p)^2$ it holds, for every $\xi \geq \zeta_D$:

$$-\frac{\xi - \delta_D}{2}\|v\|^2 \leq \langle \mathrm{Hess}\,\Phi_p(\gamma(t))[v] - \frac{\xi + \delta_D}{2}v, v\rangle \leq \frac{\xi - \delta_D}{2}\|v\|^2,$$

due to Fact 1.3. So for $\beta \in \{-1, 1\}$ we obtain the following bound:

$$-2\beta\langle D_t \operatorname{Log}_{\gamma(t)}(x), V(t)\rangle = 2\beta\langle \operatorname{Hess}\Phi_p(\gamma(t))[\gamma'(t)], V(t)\rangle$$

$$= 2\beta\langle (\ \operatorname{Hess}\Phi_p(\gamma(t)) - \frac{\xi + \delta_D}{2}I\ )[\gamma'(t)], V(t)\rangle + \beta\langle(\xi + \delta_D)\gamma'(t), V(t)\rangle$$

$$\leq 2\|\operatorname{Hess}\Phi_p(\gamma(t)) - \frac{\xi + \delta_D}{2}I\| \cdot \|\gamma'(t)\| \cdot \|V(t)\| + \beta\langle(\xi + \delta_D)\gamma'(t), V(t)\rangle$$

$$\leq 2\frac{\xi - \delta_D}{2}\|\gamma'(t)\| \cdot \|V(t)\| + \beta\langle(\xi + \delta_D)\gamma'(t), V(t)\rangle$$

$$\overset{\text{①}}{=} 2\frac{\xi - \delta_D}{2}\|z^x\| \cdot \|a + (1-t)z^x\| + \beta(\xi + \delta_D)\langle z^x, a + (1-t)z^x\rangle$$

Gauss lemma is used in the last summand of ①. Now, if $\beta = -1$, we have

$$-2\langle D_t \operatorname{Log}_{\gamma(t)}(x), V(t)\rangle \geq -2\frac{\xi - \delta_D}{2}\|z^x\| \cdot \|a + (1-t)z^x\| + (\xi + \delta_D)\langle z^x, a + (1-t)z^x\rangle$$

$$\overset{\text{①}}{\geq} -\frac{\xi - \delta_D}{2(1-t)}(\|(1-t)z^x\|^2 + \|a + (1-t)z^x\|^2) + (\xi - \delta_D)\langle z^x, a + (1-t)z^x\rangle - 2\delta_D\langle -z^x, a + (1-t)b\rangle$$

$$\geq -\frac{\xi - \delta_D}{2(1-t)}(\|a\|^2 + 2\langle a + (1-t)z^x\rangle) + (\xi - \delta_D)\langle z^x, a\rangle - 2\delta_D\langle -z^x, a + (1-t)b\rangle$$

$$\geq -\frac{\xi - \delta_D}{2(1-t)}\|a\|^2 - 2\delta_D\langle D_t V(t), V(t)\rangle. \tag{12}$$

On the other hand, analogously, if $\beta = 1$, we have

$$-2\langle D_t \operatorname{Log}_{\gamma(t)}(x), V(t)\rangle \leq 2\frac{\xi - \delta_D}{2}\|z^x\| \cdot \|a + (1-t)z^x\| + (\xi + \delta_D)\langle z^x, a + (1-t)z^x\rangle$$

$$\overset{\text{①}}{\leq} \frac{\xi - \delta_D}{2(1-t)}(\|(1-t)z^x\|^2 + \|a + (1-t)z^x\|^2) - (\xi - \delta_D)\langle z^x, a + (1-t)z^x\rangle - 2\xi\langle -z^x, a + (1-t)b\rangle$$

$$\leq \frac{\xi - \delta_D}{2(1-t)}(\|a\|^2 + 2\langle a + (1-t)z^x\rangle) - (\xi - \delta_D)\langle z^x, a\rangle - 2\xi\langle -z^x, a + (1-t)b\rangle$$

$$\leq \frac{\xi - \delta_D}{2(1-t)}\|a\|^2 - 2\xi\langle D_t V(t), V(t)\rangle, \tag{13}$$

where ① is Young's inequality $2cd \leq c^2 + d^2$. Combining (11), (12), (13), we obtain

$$-\frac{\xi - \delta_D}{2(1-t)}\|a\|^2 - 2(\delta_D - 1)\langle D_t V(t), V(t)\rangle \leq \frac{d}{dt}w(t) \leq \frac{\xi - \delta_D}{2(1-t)}\|a\|^2 - 2(\xi - 1)\langle D_t V(t), V(t)\rangle.$$

Integrating between $0$ and $r < 1$, it results in

$$\frac{\xi - \delta_D}{2}\log(1-r)\|a\|^2 - (\delta_D - 1)(\|V(r)\|^2 - \|V(0)\|^2) \leq w(r) - w(0)$$

$$\leq -\frac{\xi - \delta_D}{2}\log(1-r)\|a\|^2 - (\xi - 1)(\|V(r)\|^2 - \|V(0)\|^2).$$

Using the bound $-\log(1-r) \leq \frac{r}{1-r}$ for $r \in [0, 1)$ and using the values of $w(\cdot)$ and $V(\cdot)$, we obtain the result. □

**Corollary B.2.** *Let* $x, y, p \in \mathcal{M}$ *be the vertices of a uniquely geodesic triangle of diameter $D$, and let* $z^x \in T_x\mathcal{M}$, $z^y \overset{\text{def}}{=} \Gamma_x^y(z^x) + \operatorname{Log}_y(x)$, *such that* $y = \operatorname{Exp}_x(rz^x)$ *for some* $r \in [0, 1)$. *Then, the following holds*

$$\|z^y - \operatorname{Log}_y(p)\|^2 + (\delta_D - 1)\|z^y\|^2 \geq \|z^x - \operatorname{Log}_x(p)\|^2 + (\delta_D - 1)\|z^x\|^2,$$

*and*

$$\|z^y - \operatorname{Log}_y(p)\|^2 + (\zeta_D - 1)\|z^y\|^2 \leq \|z^x - \operatorname{Log}_x(p)\|^2 + (\zeta_D - 1)\|z^x\|^2.$$

*Proof.* Use Lemma B.1 with $a^y = 0$. Note that this corollary allows $r = 1$ as well. We obtain this result, by continuity, by taking a limit when $r \to 1$. $\square$

The following is a lemma that is already known and is used extensively in Riemannian first-order optimization. It turns out it is a special case of Corollary B.2.

**Corollary B.3 (Cosine-Law Inequalities).** *For the vertices $x, y, p \in \mathcal{M}$ of a uniquely geodesic triangle of diameter $D$, we have*

$$\langle \mathrm{Log}_x(y), \mathrm{Log}_x(p) \rangle \geq \frac{\delta_D}{2} d(x,y)^2 + \frac{1}{2} d(p,x)^2 - \frac{1}{2} d(p,y)^2.$$

*and*

$$\langle \mathrm{Log}_x(y), \mathrm{Log}_x(p) \rangle \leq \frac{\zeta_D}{2} d(x,y)^2 + \frac{1}{2} d(p,x)^2 - \frac{1}{2} d(p,y)^2$$

*Proof.* This is Corollary B.2 for $r = 1$. Indeed, given $y \in \mathcal{T}$ we can use Corollary B.2 with $z^x = \mathrm{Log}_x(y)$. Note that in such a case we have $\|z^x\| = d(x,y)$ and $z^y = 0$. Using $\|\mathrm{Log}_y(p)\| = d(y,p)$ and

$$\|z^x - \mathrm{Log}_x(p)\| = \|z^x\|^2 - \langle z^x, \mathrm{Log}_x(p) \rangle + \|\mathrm{Log}_x(p)\|^2$$
$$= d(x,y)^2 - 2\langle \mathrm{Log}_x(y), \mathrm{Log}_x(p) \rangle + d(p,x)^2,$$

we obtain the result. $\square$

**Remark B.4.** *Actually, in Hadamard manifolds, if we substitute the constants $\delta_D$ and $\zeta_D$ in the previous Corollary B.3 by the tighter constants $\delta_{d(p,x)}$ and $\zeta_{d(p,x)}$, the result also holds. See [ZS16].*

We now proceed to prove a lemma that intuitively says that solving the exact proximal point problem can be used to lower bound $f$. One should think about the following lemma as being applied to $y \leftarrow y_k^*$, $x \leftarrow x_k$. Compare the result of the following lemma with the Euclidean equality $\langle g, p - y \rangle = \langle g, p - x \rangle + \|g\|^2$, for $g = x - y$ and $x, y, p \in \mathbb{R}^n$.

**Lemma B.5.** *Let $x, y, p \in \mathcal{M}$ be the vertices of a uniquely geodesic triangle $\mathcal{T}$ of diameter $D$. Define the vectors $g \stackrel{\text{def}}{=} \mathrm{Log}_y(x) \in T_y\mathcal{M}$ and $g^x = \Gamma_y^x(g) = -\mathrm{Log}_x(y) \in T_x\mathcal{M}$. Then we have*

$$\langle g, \mathrm{Log}_y(p) \rangle \geq \langle g^x, \mathrm{Log}_x(p) \rangle + \delta_D \|g\|^2,$$

*and*

$$\langle g, \mathrm{Log}_y(p) \rangle \leq \langle g^x, \mathrm{Log}_x(p) \rangle + \zeta_D \|g\|^2.$$

*Proof (Lemma B.5).* Using the definition of $g$, we have ① below, by the first part of Corollary B.3:

$$\langle g, \mathrm{Log}_y(p) \rangle \stackrel{①}{\geq} \frac{\delta_D}{2} \|g\|^2 + \frac{d(y,p)^2}{2} - \frac{d(x,p)^2}{2}$$
$$\stackrel{②}{\geq} \langle g^x, \mathrm{Log}_x(p) \rangle + \delta_D \|g^x\|^2,$$

and in ② we used Corollary B.3 again but with a different choice of vertices so we have $\frac{d(y,p)^2}{2} \geq \frac{\delta_D}{2} \|g^x\|^2 + \frac{d(x,p)^2}{2} + \langle g^x, \mathrm{Log}_x(p) \rangle$.

The proof of the second part is analogous: using the definition of $g$, we have ① below, by the second part of Corollary B.3:

$$\langle g, \mathrm{Log}_y(p) \rangle \stackrel{①}{\leq} \frac{\zeta_D}{2} \|g\|^2 + \frac{d(y,p)^2}{2} - \frac{d(x,p)^2}{2}$$
$$\stackrel{②}{\leq} \langle g^x, \mathrm{Log}_x(p) \rangle + \zeta_D \|g^x\|^2,$$

and in ② we used Corollary B.3 again but with a different choice of vertices so we have $\frac{d(y,p)^2}{2} \leq \frac{\zeta_D}{2} \|g^x\|^2 + \frac{d(x,p)^2}{2} + \langle g^x, \mathrm{Log}_x(p) \rangle$. $\square$

 # C  Other subroutines

We provide two other subroutines that optimize functions that are $\mu$-strongly g-convex and $L$-smooth with linear rates and thus they can be used as subroutines for Line 8 in Algorithm 1. This yields accelerated algorithms for each of them.

For the first subroutine, we change the analysis but use the same algorithm as ZS16: Projected Riemannian Gradient descent $x_{t+1} \leftarrow P_X(\mathrm{Exp}_{x_t}(-\eta\nabla f(x_t)))$ but we set learning rate $\eta \stackrel{\text{def}}{=} (2 - \zeta_D)/L$. Let $\tilde{x}_{t+1} \stackrel{\text{def}}{=} \mathrm{Exp}_{x_t}(-\eta\nabla f(x_t))$. First we show the following inequality that results from applying smoothness to the first part and strong g-convexity to the second one.

$$
\begin{aligned}
0 \leq f(\tilde{x}_{t+1}) - f(x^*) &= f(\tilde{x}_{t+1}) - f(x_t) + f(x_t) - f(x^*) \\
&\leq \langle \nabla f(x_t), \tilde{x}_{t+1} - x_t \rangle + \frac{L}{2}\|\tilde{x}_{t+1} - x_t\|_{x_t}^2 + \langle \nabla f(x_t), x_t - x^* \rangle - \frac{\mu}{2}\|x_t - x^*\|_{x_t}^2 \\
&= \langle \nabla f(x_t), \tilde{x}_{t+1} - x^* \rangle + \frac{L\eta^2}{2}\|\nabla f(x_t)\|^2 - \frac{\mu}{2}\|x_t - x^*\|_{x_t}^2 \\
&= \langle \nabla f(x_t), x_t - x^* \rangle + \left(\frac{L\eta^2}{2} - \eta\right)\|\nabla f(x_t)\|^2 - \frac{\mu}{2}\|x_t - x^*\|_{x_t}^2.
\end{aligned}
\tag{14}
$$

Now, we have the following bound, bounding the distance to the minimizer, from which we will derive convergence rates for projected RGD:

$$
\begin{aligned}
d(\tilde{x}_{t+1}, x^*)^2 &\stackrel{\text{①}}{\leq} (\zeta - 1)\eta^2\|\nabla f(x_t)\|^2 + \|x^* - \tilde{x}_{t+1}\|_{x_t}^2 \\
&\stackrel{\text{②}}{\leq} \|x^* - x_t\|_{x_t}^2 + 2\eta\langle \nabla f(x_t), x^* - x_t \rangle + \zeta\eta^2\|\nabla f(x_t)\|^2 \\
&\stackrel{\text{③}}{\leq} \left(2\eta - \frac{\zeta\eta}{1 - \frac{L\eta}{2}}\right)\langle \nabla f(x_t), x^* - x_t \rangle + \left(1 - \frac{\mu\zeta\eta}{1 - \frac{L\eta}{2}}\right)\|x^* - x_t\|_{x_t}^2.
\end{aligned}
\tag{15}
$$

where in ① we used the Euclidean cosine theorem along with Corollary B.3. Inequality ② develops the square $\|x^* - \tilde{x}_{t+1}\|_{x_t}^2 = \|x^* - x_k - \eta\nabla f(x_t)\|_{x_t}^2$ and ③ uses (14), where the inequality has been multiplied by $-\zeta\eta^2(L\eta^2/2 - \eta)^{-1} = \frac{\zeta\eta}{1 - \frac{L\eta}{2}}$ ($\geq 0$, since we assume $\eta \in [0, 2/L)$) in both sides.

Now, since $\langle \nabla f(x_t), x^* - x_t \rangle \leq 0$, we want to make the factor alongside it be $\geq 0$ in order to drop it. That means, it should be $2\eta - \frac{\zeta\eta}{1 - \frac{L\eta}{2}} \geq 0$ which is equivalent to $\eta \leq \frac{2-\zeta}{L}$. By setting $\eta$ exactly to the value $\frac{2-\zeta}{L}$ and assuming $\zeta < 2$, we have $\frac{\zeta\eta}{1 - \frac{L\eta}{2}} = 2(2 - \zeta)/L$ and so we can conclude:

$$
d(x_{t+1}, x^*)^2 \leq d(\tilde{x}_{t+1}, x^*)^2 \leq \left(1 - \frac{2\mu(2 - \zeta)}{L}\right)d(x_t, x^*)^2.
$$

which is linear convergence, as desired.

For the second subroutine, we assume access to the operation

$$
x_{t+1} = \underset{y \in \mathcal{X}}{\arg\min}\{\langle \nabla f(x_t), y - x_t \rangle_{x_t} + \frac{L}{2}d(x_t, y)^2\},
$$

and define the algorithm as the sequential application of it. This subproblem, in the Euclidean case, is equivalent to the projection operator of $\tilde{x}_{t+1} = \mathrm{Exp}_{x_t}(-\eta\nabla f(x_t))$. However, in the Riemannian case, this and the metric-projection operator $P_{\mathcal{X}}(x_{t+1})$ are two different things in general. Define the notation $\phi(x) \stackrel{\text{def}}{=} (f + I_{\mathcal{X}})(x)$. Then, we have

$$\phi(x_{t+1}) \overset{①}{\leq} m_L(x_t, x_{t+1})$$

$$= \min_{x \in \mathcal{M}} \left\{ f(x_t) + \langle \nabla f(x_t), x - x_t \rangle_{x_t} + \frac{L}{2} d(x, x_t)^2 + I_{\mathcal{X}}(x) \right\}$$

$$\overset{②}{\leq} \min_{x \in \mathcal{M}} \left\{ f(x) + \frac{L}{2} d(x, x_t)^2 + I_{\mathcal{X}}(x) \right\}$$

$$= \min_{x \in \mathcal{M}} \left\{ \phi(x) + \frac{L}{2} d(x, x_t)^2 \right\}$$

$$\overset{③}{\leq} \min_{\alpha \in [0,1]} \left\{ \alpha \phi(x^*) + (1 - \alpha) \phi(x_t) + \frac{L \alpha^2}{2} d(x^*, x_t)^2 \right\}$$

$$\overset{④}{\leq} \min_{\alpha \in [0,1]} \left\{ \phi(x_t) - \alpha \left( 1 - \alpha \frac{L}{\mu} \right) (\phi(x_t) - \phi(x^*)) \right\}$$

$$\overset{⑤}{=} \phi(x_t) - \frac{\mu}{2L} (\phi(x_t) - \phi(x^*)).$$

Above, ① holds by smoothness and ② holds by g-convexity of $f$ (I thought maybe using strong convexity one can improve but it is not by much, it results in convergence rates of $O((\frac{L}{\mu} - 1) \log(1/\varepsilon)$ instead of $O(\frac{L}{\mu} \log(1/\varepsilon)$. So I am not using it). Inequality ③ results from restricting the minimum to the geodesic segment between $x^*$ and $x_t$ and uses g-convexity of $\psi$. In ④, we used strong convexity of $\phi$ to bound $\frac{\mu}{2} d(x^*, y_k)^2 \leq \phi(x_t) - \phi(x^*)$. Finally, in ⑤ we substituted $\alpha$ by the value that minimizes the expression, which is $\mu/2L$.

Subtracting $\phi(x^*)$ to the inequality above, we obtain $\phi(x_{t+1}) - \phi(x^*) \leq \left(1 - \frac{\mu}{2L}\right) (\phi(x_t) - \phi(x^*))$. As we wanted to prove, there is linear convergence.