# OpenReview forum: "Metric-Projected Accelerated Riemannian Optimization: Handling Constraints to Bound Geometric Penalties"
_NeurIPS.cc/2022/Conference — NeurIPS 2022 Submitted_

### Official Review · Reviewer_9V2i · 2022-07-08

**Rating:** 6
**Confidence:** 4
**Soundness:** 3 good
**Presentation:** 3 good
**Contribution:** 3 good

**Summary:**

This paper proposed an accelerated first-order optimization algorithm for smooth and
(strongly) geodesically-convex functions over a compact and geodesically-convex set
on Hadamard manifolds. It is proven in this paper that this Riemannian algorithm
enjoys the same convergence rate as the Euclidean AGD algorithm without convex constraints.
Though there exist some relavent works, none of them fully generalize the Euclidean AGD
convergence results to the Riemannian setting.

**Questions:**

The requirements of a geodesically convex function over a geodesically-convex set on Hadamard manifolds
are quite strong and already limit the usage for many important applications. The author gives two manifolds in
the paper, one is a hyperbolic manifold and the other one is the manifold of SPD manifold. Giving a few applications
would be helpful.

The metric projection oracle seems not easily available. Giving a few examples and potentials in applications
would be useful.

The algorithm needs the value of \zeta_{2D}. How do users get this value?

**Limitations:**

I think the authors need to comment on the implementation and numerical performance of this proposed algorithm.

**Strengths And Weaknesses:**

The contribution of this paper is on the theoretical aspect. It is interesting in the sense
that it firstly gives an algorithm that fully extends the Euclidean AGD to the Riemannian setting
with constraints. However, numerically, this algorithm is hard to use. It involves quite a
few unknown operators and parameters, and expensive operators.

---

> ### Author Response · Authors · 2022-07-31
> **reply**
>
> Thank you for your time and your review. Please check as well the announcement we wrote regarding an error in previous work and our new subroutines that we design in substitution. Also, we are happy to announce that we have a new result regarding a cheap metric-projected oracle that works in the case where most applications lie (no further in-manifold constraints so we can pick a ball constraint to bound geometric penalties).
>
> > It is proven in this paper that this Riemannian algorithm enjoys the same convergence rate as the Euclidean AGD algorithm **without convex constraints**.
>
> We would like to note that we compare our algorithm with constrained Euclidean accelerated gradient descent (see the clarification of footnote 1 in page 4), which supports a generic projection oracle and calls it once per gradient oracle call, as we do.
>
>
> > However, numerically, this algorithm is hard to use.
>
> See the metric-projection oracle we describe on the announcement to all reviewers for a very important case in which the projection oracle can be readily computed, which is a new result we have. In general, the particular constrained problem at hand requires the user to implement the projection oracle, **as it is the case with accelerated gradient descent in the Euclidean case**. But virtually all Riemannian optimization problems we know have no initial in-manifold constraints, and in that case our ball constraints are enough to bound geometric penalties and we can now implement the projection oracle easily.
>
>
> > The requirements of a geodesically convex function over a geodesically-convex set on Hadamard manifolds are quite strong and already limit the usage for many important applications. The author gives two manifolds in the paper, one is a hyperbolic manifold and the other one is the manifold of SPD manifold. Giving a few applications would be helpful.
>
> We argue that the requirements for the function and the set are not strong.
>
> On the one hand, Riemannian optimization is of great interest because many problems have been found to be geodesically convex, and it has recently been intensively researched because of this fact. Operator scaling, Gaussian mixture models... and many others are geodesically convex problems. See [HS20] for more examples, as we cited in line 53. We will emphasize this. On the other hand, in Euclidean optimization, the deep understanding of convex optimization has been essential in order to design efficient (or even optimal) algorithms for some tasks in non-convex optimization, such as approximating critical points or approximate local minima. It is often the case that convex subroutines are used in these algorithms. We expect a similar thing to hold for Riemannian optimization which further motivates the study of problems that are geodesically convex.
>
> The geodesic convexity of the feasible set is not strong either. For instance, there is a natural set that is geodesically convex and uniquely geodesic: a ball $\{x | d(x, x_0) < R \}$ for a center $x_0$, which is enough for most applications. But beyond this fact, our analysis is general enough to support any geodesically convex constraints for which one can implement a metric projection oracle. We will make these insights more explicit in the final version of the paper.
>
> > The metric projection oracle seems not easily available. Giving a few examples and potentials in applications would be useful.
>
> We have a new result regarding this, see the announcement to all reviewers, whose content we will add to the final version.
>
>
> > The algorithm needs the value of \zeta_{2D}. How do users get this value?
>
> Given the feasible set one can compute its diameter D. The manifolds these kinds of methods are applied to are well studied and bounds on the sectional curvature are known. Given these quantities, one can easily compute $\zeta_{2D}$ by using its definition at the end of page 3: $2D\sqrt{|\kappa_{\min}|} \coth(2D\sqrt{|\kappa_{\min}|})$.
>
> > I think the authors need to comment on the implementation and numerical performance of this proposed algorithm.
>
> We will comment on this and on the new easy projection oracle. Reviewer qP9H also asked about the implementation and its practicality and we can add a remark with the content of that reply, with the three computational requirements (A),(B),(C) we listed to Reviewer qP9H and their implementability.
>
> We hope to have clarified your questions and we would be happy to further answer any other questions that may arise.

---

> ### Author Response · Authors · 2022-08-09
> **our rebuttal?**
>
> Dear Reviewer 9V2i,
>
> We hope to have convinced you that the assumptions on the function and feasible set are not strong and that they are the right thing to use, which is the reason why they are startanrd. We hope to have been clear with respect to our extra result regarding the projection oracle in the Riemannian unconstrained case (the most important case), now we can project cheaply, resulting in a practical algorithm that only requires a gradient oracle and standard geometric quantities. Moreover, we provide new subroutines for unaccelerated linear convergence of constrainted strongly g-convex problems, given that we found that the one in the COLT paper of Zhang and Sra was wrong.
>
> We did not hear from you yet, but we would be delighted to learn whether our responses answered you questions. Thanks.

---

### Official Review · Reviewer_qP9H · 2022-07-11

**Rating:** 5
**Confidence:** 3
**Soundness:** 3 good
**Presentation:** 3 good
**Contribution:** 3 good

**Summary:**

The paper proposes accelerated first-order methods for optimizing smooth and geodesically convex functions on Hadamard manifolds.

**Questions:**

see above

**Limitations:**

see above

**Strengths And Weaknesses:**

Acceleration on Riemannian manifolds is an important and timely topic that has recently received a lot of interest in the optimization community. The literature review is comprehensive and the background section is well-written. The theoretical results are well-motivated and presented.

My main concern is the metric projection oracle and the resulting comparison results with existing methods. If I understand correctly, the complexity results are given with respect to the number of queries to the metric projection oracle. However, the cost of one such query is not clear to me from the text (also not from Rmk. 2.3). If you factor in this cost, does the method still compare favorably to prior methods? Please clarify, if I am misunderstanding.

How practical is the proposed method? Can Alg. 1 (including the subproblems) be implemented efficiently for at least some classical problems, e.g., some of the examples listed in the first paragraph of the introduction?

---

> ### Author Response · Authors · 2022-07-31
> **reply**
>
> Thank you for your review. Please see as well the post we wrote addressing all reviewers regarding an error in previous work and our new subroutines and our new metric-projection oracle.
>
> > My main concern is the metric projection oracle and the resulting comparison results with existing methods
>
> The analysis provides bounds on the number of queries to gradient and metric projection oracles. This is exactly the same as what one gets in accelerated constrained Euclidean convex optimization. As with Euclidean optimization, some projection oracles can be more expensive and some are cheaper. That being said, now we have a way to implement very cheaply a metric-projection oracle when the constraints consist of a ball, and this case encompasses most problems, as most problems do not have in-manifold constraints and even though we need to impose some constraints in order to bound geometric penalties, we can decide to impose the constraint we want, and it can be a ball constraint.
>
> > If you factor in this cost, does the method still compare favorably to prior methods? Please clarify, if I am misunderstanding.
>
> The method does compare favorably to prior methods regardless of the cost of the metric projection oracle. The main reason is that most current methods make an undesirable assumption which is not known how to satisfy or it is not even clear if it can be satisfied (that the iterates will stay in some pre-specified bounded set without any mechanism to enforce this), so they do not provide a full implementable algorithm for which one can give guarantees. The only two methods that allow to enforce constraints are in [Marr22] and [CB21] (see table) and they work with a ball constraint and different ball projections than what we use but equally cheap. But these two methods only apply to limited settings: the former to constant curvature and the latter to local optimization and only in strongly g-convex problems. Besides from the possibility of implementing a cheap ball projection oracle, the theory of our method allows anyone to have an accelerated method if they have more complex constraints and they implement a projection oracle (exactly as in Euclidean constrained convex optimization). Again, we emphasize that the ball constraint encompasses virtually all Riemannian problems we are aware of, as unconstrained Riemannian accelerated optimization requires constraints as well in order to bound geometric penalties and we can pick a simple one (a ball) in order to do so.
>
> > How practical is the proposed method? Can Alg. 1 (including the subproblems) be implemented efficiently for at least some classical problems, e.g., some of the examples listed in the first paragraph of the introduction?
>
> The algorithm requires (A) computation of geometric operations (exponential map, inverse exponential map and parallel transport) (B) gradient oracle, (C) metric projection oracle / subroutine. (A) can be computed for all the applications listed and cited in the introduction, they are basic operations that previous Riemannian optimization algorithms for these problems use as well. That being said, optimization with orthogonality constraints and sparse PCA require manifolds that are not Hadamard. The rest of the applications use Hadamard manifolds. As we said in the conclusion of the paper, extending the analysis to manifolds with positive curvature is an interesting future direction of research. (B) can be readily computed using automatic differentiation and access to the parametrization of the manifold. As for (C) as explained above for the almost universal case in which there are no in-manifold constraints we can now implement this cheaply. In the general case, efficiency of this operation depends on the constraints (as in the Euclidean case and as in analysis like projected gradient descent or constrained accelerated gradient descent).
>
>
> We hope to have clarified all of your questions. In sum, our intention with this work was to achieve acceleration in the Riemannian setting with an algorithm that is implementable and comes with guarantees, by removing the undesirable assumption most works had (it is not realistic the iterates stay in a pre-specified set if there are no mechanisms to enforce this, so the guarantees of these algorithms are not usable) and by obtaining much more general results than the two works that did not made the assumption. We note our analysis is general enough that it can work with a general metric projection for their problems, as in constrained Euclidean convex optimization. We hope that this level of generality allows for new research whether it is on the implementation of some other projection operators or some other linearly convergent algorithms that can be used as subroutines. Additionally, our new result allows us to implement the projection oracle in a very broad case, making the algorithm efficient and practical.

---

> > ### Comment · Reviewer_qP9H · 2022-08-09
> > **Response to author reply**
> >
> > Thanks to the authors for their comments. After reading your response, I still think that this is a borderline paper (leaning towards acceptance); hence, I will keep my score.

---

> > > ### Author Response · Authors · 2022-08-09
> > > **Re: Response to author reply**
> > >
> > > Could you clarify why after the response your impression of the paper is the same? The main concerns, namely the implementation of the metric projection oracle, how the method compares favorably to other methods if one takes into account the projection oracle, and the practicality of the method were all addressed:
> > >
> > > For all the previously known problems there are no in-manifold constraints and since we need to impose at least some constraint to bound geometric penalties we can decide to impose a Riemannian ball as constraints, whose projection oracle is given in closed form. Thus, the method only requires a gradient oracle and the standard geometric operations of the exponential map, inverse exponential map and parallel transport.
> > >
> > > If there are any other concerns please could you clarify so we can comment on them? Thank you.

---

### Official Review · Reviewer_HxJc · 2022-07-11

**Rating:** 5
**Confidence:** 3
**Soundness:** 3 good
**Presentation:** 4 excellent
**Contribution:** 3 good

**Summary:**

The authors present "an accelerated first-order method for the optimization of smooth and (strongly or not) geodesically-convex functions over a compact and geodesically convex set in Hadamard manifolds, that we access to via a metric-projection oracle". For this oracle, they use "projected Riemannian gradient descent to implement an inexact proximal point operator".
The analysis aims to establish 'an accelerated rate' up to multiplicative factors induced by the manifold's sectional curvature (Line 166, Fact 1.3, Lines 149-155), as well as some log factors (Remark A.5, Line 838, Remark 2.3). An extensive list of recent work on Riemannian optimization has been provided.


**Questions:**

It would be helpful to provide a remark, specifically gathering all of the "universal constants" and "logarithmic factors" (e.g., Remark A.5, Line 838, Remark 2.3, etc); and/or the source of their introduction into the analysis.

Footnote 0: some of the links seem to be broken; for example, for \kappa_\min.

There is an impossibility argument on Lines 246-250 that I did not follow. Could you please elaborate?

A short discussion on the gains from the 'approximate' computation (Remark 2.3), in some of the Riemannian optimization instances, could be helpful.

Some examples/references for the note on Lines 365-367 could be useful. ("We note that any other algorithm with linear convergence rates for constrained strongly g-convex, smooth problems that works with a metric-projection oracle can be used as a subroutine to obtain an accelerated Riemannian algorithm.")

Could the authors elaborate on
- "due to fundamental properties of their methods" on Line 63;
- "which is not amenable to optimization" on Line 73-74;

In relation to "The Riemannian proximal point subroutine we design is of independent interest", on Lines 79-80 (and in the abstract), it might be beneficial to present this subroutine in an algorithmic form instead of plain text.


Are there any limitations of the presented algorithm or analysis that the authors feel could inform a better comparison with the existing literature as well as future work? Including possibilities of alternatives in certain aspects. Other examples: any further details regarding the multiplicative factors (Line 382), or any indication or difficulty in extending to other manifolds (Lines 383-384)?


**Limitations:**

Please see previous comments on the discussion of technical contributions as well as on logarithmic factors.


**Strengths And Weaknesses:**

The writing and the flow are excellent but only provide a very high level idea of the work and contributions of the current paper. A considerable part of the paper, up until the middle of page 7, out of 9 pages, focuses on review of background material, review of previous works, and "insights", and the technical statements and proofs have been presented in the appendices. I would also leave a proper assessment of the claims to reviewers with close familiarity to those recent works cited upon which the paper claims advantages. As an example a lot of the background reviewed on Lines 85-152 remains unused within the main paper.

---

> ### Author Response · Authors · 2022-07-31
> **reply**
>
> > In relation to "The Riemannian proximal point subroutine we design is of independent interest", on Lines 79-80 (and in the abstract), it might be beneficial to present this subroutine in an algorithmic form instead of plain text.
>
> The main points are the warm start which is just one step, invoking an unaccelerated linearly converging algorithm and the analysis behind it. We can add an {algorithm} environment summarizing this.
>
> > Are there any limitations of the presented algorithm or analysis that the authors feel could inform (A) a better comparison with the existing literature as well as (B) future work? Including possibilities of alternatives in certain aspects. Other examples: any further details regarding the multiplicative factors (Line 382), or any indication or difficulty in extending to other manifolds (Lines 383-384)?
>
> (A) Besides from the now new thing implied by our discovery of the analysis of (Zhang and Sra COLT 2016) being wrong and thus us needing to now use other subroutines, that we have designed (see announcement to all reviewers), which we will add to the final version and comment on, we think we have pointed out all limitations and comparisons extensively. (B) Regarding future work, there is the possibility that the $\log(1/epsilon)$ could be removed by exploring how to adapt recent techniques on Euclidean convex optimization that can use accelerated proximal point methods with a constant number of iterations in the subroutine, like "Contracting proximal methods for smooth convex optimization", but it is not clear how to achieve something similar in the Riemannian case. We currently do not know how one could reduce the geometric factors, but the fact that [KY22] had $\zeta$ instead of $\zeta^2$, even if they work under more stringent assumptions, could mean that another algorithm could improve on that aspect as well.

---

> ### Author Response · Authors · 2022-07-31
> **reply**
>
> Thank you for your review, your time and your many questions! Please check as well the post directed to all reviewers that we wrote. We reply to your comments here below and hope that you find them useful and they clarify the main points of our work.
>
> > The writing and the flow are excellent but only provide a very high level idea of the work and contributions of the current paper. (...) As an example a lot of the background reviewed on Lines 85-152 remains unused within the main paper.
>
> According to our experience, reviewers and readers usually ask us about a preliminaries section in order to understand the concepts that follow with algorithms of this kind, specially when they are in a setting that is not very standard and well known, as it is the case with Riemannian optimization. That said, we believe all concepts in lines 85-152 are necessary to read the 9 pages long main part of the paper: to understand the insights explained in high-level in section 1 and then compared with previous work, to understand the pseudocode, and to understand the technical discussion in section 2. All the other technical details are provided in the appendix.
>
> > It would be helpful to provide a remark, specifically gathering all of the "universal constants" and "logarithmic factors" (e.g., Remark A.5, Line 838, Remark 2.3, etc); and/or the source of their introduction into the analysis.
>
> We can provide the source of the introduction of our constants and logarithmic factors for clarifying purposes in the final version of the paper.
>
> > Footnote 0: some of the links seem to be broken; for example, for \kappa_\min.
>
> Thanks for pointing this out, we have fixed that link and some others. We have just realized we can output warnings for the links whose target we forgot to specify and have now fixed them all.
>
> > There is an impossibility argument on Lines 246-250 that I did not follow. Could you please elaborate?
>
> It was not exactly an impossibility argument. In some problems (like for strongly g-convex problems) one can argue that reducing the function value implies the iterates get closer to the minimizer and so a monotone method has bounded iterates. Or it could be that in some setting one can prove that an algorithm with some particular learning rates will have bounded iterates. One could think in such a case that the constraints that we use to bound geometric penalties are not needed because the algorithms naturally stay bounded. But we point out this is not enough to guarantee bounded geometric penalties because given a set of diameter D where we want the iterates to remain, algorithms use the value D to set learning rates, and provide convergence rates. The existence of boundedness of the iterates could only guarantee boundedness with diameter D' > D deeming the analysis unusable, unless we change the learning rates to depend on D', in which case the boundedness of the iterates could have even greater diameter D'' and so on.
>
>
> > Some examples/references for the note on Lines 365-367 could be useful. ("We note that any other algorithm with linear convergence rates for constrained strongly g-convex, smooth problems that works with a metric-projection oracle can be used as a subroutine to obtain an accelerated Riemannian algorithm.")
>
> Here we were pointing out that our results consist of a general reduction and future algorithms with linear rates can be used as a subroutine for obtaining an accelerated algorithm. This is very interesting because if, for instance, someone were to show **unaccelerated** linear rates for constrained strongly g-convex finite-sum problems, then our results provide an **accelerated** algorithm for g-convex (strongly or not) for the finite-sum case. Such algorithms are not known. And similarly for other settings. We can change the sentence to make clear we are not aware of any other linearly convergence algorithms besides from the two subroutines we provide (see the announcement we made to all reviewers) and we can add the example made in this paragraph.
>
> > Could the authors elaborate on "due to fundamental properties of their methods" on Line 63;
>
> We were referring to the following: their algorithms and analysis need to advance against the direction of the gradient for some length, which is incompatible with constraints or projections, and it is a fundamental part of their methods. We will change this sentence to be more precise.
>
> > elaborate on "which is not amenable to optimization" on Line 73-74;
>
> We will make the sentence more clear, we were just emphasizing the hardness of the problem being non-convex.

---

> ### Author Response · Authors · 2022-08-09
> **Does our rebuttal address all your concerns?**
>
> Dear reviewer HxJc
>
> Please can you let us know what you think? We believe our work improves greatly over the previous algorithms attempting to achieve Riemannian acceleration and we make other important technical contributions (such us accelerated Riemannian proximal point method with exact prox, not needing a point with 0 gradient to be inside of the set, and a generic accelerating reduction)
>
> We believe we replied to all of your concerns and we hope to having giving you a more comprehensive view of our work. Do our responses indeed address all the concerns? If so, please let us know! If not, please comment so we can clarify further.

---

> > ### Comment · Reviewer_HxJc · 2022-08-09
> > **response to author reply**
> >
> > Thank you for further elaborations. I have read other reviews and discussions as well. My main concern is still that the provided discussions are mostly high-level, with insufficient 'exact' references/arguments for claims (most methods having some property, most problems fitting into this framework, cheaply [which is vague if no specific application is under consideration], etc). I will keep my previous score.

---

> > > ### Author Response · Authors · 2022-08-09
> > > **Re: response to author reply**
> > >
> > > We apologize if the reviewer felt that the discussion was being too high-level, but we are happy this issue was raised so we actually have a two-way discussion and we can clarify!
> > >
> > > Please read below for a clarification on all these issues.
> > >
> > > Regarding properties of previous methods, we were always very explicit on what methods assumed the iterates to remain in a pre-specified compact set: all of them except [Mar22] and [CB21], which are limited to constant curvature and local optimization, respectively, as discussed in the paper. We cited all of the papers and provided an explanatory table indicating which methods can and which cannot deal with constraints (column "C?") Our method
> > >
> > > Regarding the "cheap" implementation of the projection oracle (its the only case in which we have referred to anything as cheap), we believe we were very explicit on what the operation is, but we apologize if the explanation was not clear and will repeat the argument here: in the post addressed to all reviewers, we proved that the metric projection if the feasible set is a Riemannian ball has a closed form! That is, there is no need to implement the projection oracle with a possibly expensive optimization subalgorithm. This projection takes the form $P_X(x) = Exp_{x_0}(R \cdot Log_{x_0}(x)/|Log_{x_0}(x)|)$ if $x\not\in X$ and $P_X(x) = x$ otherwise.  This closed form uses just one exponential map $Exp_{x_0}$ and one inverse exponential map $Log_{x_0}$, which are basic building blocks (see Algorithm 1 for more uses of these maps).
> > >
> > >
> > > Regarding the "most problems fitting into this framework [i.e. Riemannian optimization with no in-manifold constraints]" we note that all the problems that were cited that apply to our method (i.e. Hadamard manifolds and geodesically convex or strongly geodesically convex functions) have no constraints (except for the manifold, which can be considered a constrained that has been lifted by optimizing over it as the domain. Having no other constraints is what we refer to as no in-manifold constraints). These are:
> > > + Dictionary learning [CS17; SQW17]
> > > + Robust covariance estimation in Gaussian distributions [Wie12]
> > > + Gaussian Mixture Models [HS15].
> > > + Operator scaling [All+18].
> > > + One can see more problems in the survey that we cited [HS20, Section 6 "Example applications"] (Karcher mean, Wasserstein Barycenters).
> > >
> > > Thus, for all the problems above, we can decide on the constrain to use to keep the iterates of the algorithm bounded and to bound geometric penalties. And hence we can use a Riemannian ball, for which we now have a closed form solution for its metric-projection oracle  . As we said in our response to reviewer qP9H, our algorithm does not apply to optimization under orthogonality constraints and sparse PCA since those cases are defined over manifolds that are not Hadamard and extending our results to allow for positive curvature is an interesting direction of future research.
> > >
> > > Because of these applications and because of its level of generality, obtaining a Riemannian version of accelerated gradient descent to Riemannian manifolds is a problem that has received a lot of attention in the last few years [Liu+17, ZS18, Ali+19, HW19a, Ali+20, Lin+20, AS20, CB21, Mar22, KY22] plus papers studying lower bounds [HM21, CB21]. We managed to generalized the Euclidean Accelerated Gradient Descent algorithm to a practical algorithm without working with unreasonable assumptions (like assuming that the iterates will stay within some pre-specified set without imposing any projection mechanism) and improved on several other aspects. We believe this is a significant contribution to the topic.
> > >
> > > Also, for completeness, we updated our submission of the supplementary material and have added Appendix C, that contains the two new subroutines and their analyses.
> > >
> > > We hope to have been completely clear about everything that the reviewer suggested could seem it was not "exact" or too "high-level" and if that is the case we encourage to reconsider their evaluation of our work.

---

### Author Response · Authors · 2022-07-31
**Announcement for all reviewers**

TL;DR:

1. We now have a simple cheap-to-compute projection operator for a Riemannian ball constraint. In order to bound geometric penalties for problems with no in-manifold constraints (and most problems we are aware of are of this kind) we can decide to impose a ball as our constraint, so this covers most problems.

2. We very recently found that the analysis of the projected Riemannian Gradient Descent algorithm made by Zhang and Sra in their COLT 2016 paper is wrong and so we cannot use it as subroutine. Our method admits any linearly convergent algorithm for strongly g-convex smooth problems as subroutine and we designed 2 new subroutines that work under different assumptions which still allow to obtain acceleration and improve over previous works at the expense of some generality. We will add this to the final version of the paper. See below for a full discussion.

**Detailed explanation:**

**1. Ball constraint**

+ If the Riemannian problem is unconstrained (has no in-manifold constraints) we still need **some** constraint to bound geometric penalties, but **any** constraint works and we can decide to use a Riemannian ball with center at $x_0$ as constraint. Usually Riemannian optimization turns constrained problems into unconstrained ones whose domain is the manifold. That is, for most Riemannian problems (like the ones we cited) the constraints are codified by the manifold and there are no other in-manifold constraints, so a metric-projection oracle for a ball would cover most problems in Riemannian optimization.
+ We have a new, simple, but powerful result that allows us to build a very cheap projection oracle when the constraint is a ball: The metric projection of a point $x$ outside of the Riemannian ball is the point of the border of the ball that is in the geodesic that joins $x$ and $x_0$. That is, if we call $R$ the radius of the ball, we have that $P_X(x) = Exp_{x_0}(R\cdot Log_{x_0}(x)/\|Log_{x_0}(x)\|)$ if $x \not\in X$ and $P_X(x) = x$ otherwise. The proof is simple and as follows. We work with uniquely geodesic feasible sets (and a Riemannian ball in a Hadamard manifold is uniquely geodesic) and so a curve between two points is a geodesic if and only if it is globally distance-minimizing. By definition, the projection point $P_X(x)$ satisfies $d(x, P_X(x)) \leq d(x, y)$ for all $y \in X$, so if $x$ is outside of the ball then $P_X(x)$ is on the border of the ball and $d(x_0, P_X(x))=R$ and so the two geodesic segments from $x$ to $P_X(x)$ and from $P_X(x)$ to $x_0$ form a curve that is globally distance minimizing and thus this curve is the geodesic joining $x$ and $x_0$. Q.E.D.

**2. New subroutines**

The main point of our paper is to show that our Algorithm 1 and its analysis provide a generic framework for obtaining acceleration in Riemannian manifolds without making the undesirable assumption that is that the iterates remain in some pre-specified compact set. This is a generic reduction that, from a linearly convergent **unaccelerated** constrained algorithm for strongly g-convex problems, returns **accelerated** algorithms for smooth and g-convex or strongly g-convex problems.

We very recently, for the paper by Zhang and Sra "First-order Methods for Geodesically Convex Optimization" published in COLT 2016 and that contains a metric-projected Riemannian gradient descent and a "proof" of constrained linear convergence for strongly g-convex smooth problems, found out a mistake in the proof of convergence, when the problem is constrained. So we cannot use it in our Remark 2.3 to provide the example showing the reduction being used. However, our reduction is general enough that any other linearly convergent algorithm can be used as a subroutine.

There was no other such algorithm in the literature, but we have a proof of the linear convergence of two constrained Riemannian Gradient Descent algorithms for the strongly g-convex smooth setting:

+ For $\zeta < $ a function-independent universal constant and a metric-projection oracle: we have a very different analysis that shows that the same algorithm: metric-projected Riemannian Gradient Descent, enjoys unaccelerated linear convergence rates and therefore it can be used in our algorithm as a subroutine and we obtain our claimed final convergence.

---

> ### Author Response · Authors · 2022-07-31
> **.**
>
> + For **any** $\zeta$, assuming access to an algorithm that minimizes the upper bound that smoothness yields when computing $\nabla f(x_t)$ i.e. $\arg\ \min_{x\in X}\{\langle \nabla f(x_t), \operatorname{Log}_{x_t}(x) \rangle + \frac{L}{2}d(x_t, x)^2 \}$, then we have an analysis showing linear rates of convergence (and actually an approximate solution of such subproblem also works). This subproblem, in the Euclidean case, is equivalent to the projection operator. However, in the Riemannian case, this and the metric-projection operator are two different things.
>     + Solving the subproblem does not require calling the gradient oracle and therefore we show that the gradient oracle complexity of optimizing smooth and (possibly strongly) g-convex, for any $\zeta$ is as it is displayed on the table, without making any assumptions about "the iterates of the algorithm staying in a pre-specified compact set, without enforcing this condition". This is useful, among other things, because it shows that it is not possible to prove a lower bound on the oracle complexity that has a hardness of the geometry factor worse than our $\zeta^2$ (the only global constrained solution that existed, [Mar21], had an exponential penalty on $\zeta$ and it was not clear before this paper if one could have lower oracle complexity, due to the exponentially-growing volume of balls in spaces of negative curvature).
>
> We can write the analysis of these two subroutines here upon request of the reviewers, and in any case we will add them to the final version of our paper with all level of detail and we will modify the exposition of the paper accordingly (i.e., modifying the remark about the implemented subroutine, adding the new 2 different subroutines, and making clear in the text what the extent of the results is).
>
> **Conclusion**
>
> In conclusion, all of our main results still hold, except for the lower level of generality in the first point and we add a new result regarding the metric projection, that makes the method be practical:
>
> + We still solve the open question of Kim and Yang 2022, now published on ICML 2022, asking about whether acceleration can be obtained without making the assumption "the iterates of the algorithm stay in a pre-specified compact set without enforcing them to be in it". **Without making any of those assumptions:**
>     + For $\zeta <$ a universal constant, we show that projected Riemannian Gradient Descent enjoys linear rates and thus it can be used as a subroutine to yield our accelerated algorithm.
>     + For any $\zeta$, we show the upper bound in oracle complexity that enjoys the same rates as Euclidean Nesterov's accelerated gradient descent up to the geometric constant factor $\zeta^2$.
>
> + We provide a generic and cheap to compute metric-projected oracle when the constraint is a Riemannian ball. We argue that most Riemannian problems have no in-manifold constraints and therefore we can choose the constraint we like in order to bound geometric penalties and we can use a Riemannian ball. This makes the method only require a gradient oracle. Thus, it is efficient, and works under practical assumptions
>
> + If we can compute the exact Riemannian proximal point operator and we use it as the implicit gradient descent step in Line 8 of Algorithm 1, then the method is an accelerated proximal point method. One such Riemannian algorithm was previously unknown in the literature as well.
>
> + We provide the generic accelerating reduction. If a new constrained algorithm with linear rates is designed for any setting (e.g. linear convergence for constrained strongly g-convex finite-sum problems) then our framework provides an accelerated algorithm. (in the example, it would be the first accelerated Riemannian algorithm for finite-sum problems). Also if another subroutine for strongly g-convex and smooth problems is designed, besides from our two subroutines (for example, one for a generic $\zeta$) then our reduction yields an accelerated algorithm under the same setting.
>
> + Other technical details and improvements: smoothness and g-convexity is only required inside of X, we optimize with respect to a global minimizer in X without assuming this is a point with 0 gradient.

---

> > ### Author Response · Authors · 2022-08-09
> > **.**
> >
> > To all reviewers, please note we updated the supplementary material to include a new section in the Appendix (section C) including the new subroutines and their proofs of linear convergence

---

### Meta-Review · Area_Chair_x4bR · 2022-08-24

**Recommendation:** Reject
**Confidence:** Certain

**Metareview:**

The paper deals with accelerated methods on Riemannian manifolds. A particular challenge that the paper tries to address, which the AC believes is important, is related to the bounding of the iterates. The paper starts with an explicit bounding constraint on the manifold (and relaxes to the ball constraint for certain manifolds) and shows that the proposed algorithm can respect that while achieving acceleration. The reviewers including this AC see the merits of the paper. However, the paper in its current form is far from complete. A particular concern is on the empirical performance of the algorithm, resolving which should strengthen the paper. I would encourage the authors to build on the discussions and polish the paper accordingly.

Even though the paper has positive scores, the paper in its current form is a borderline paper with significant scope for improvement. To this end, the AC cannot accept the paper.

**Award:**

No

---

### Decision · Program_Chairs · 2022-09-14

Reject